# A salvage pathway maintains highly functional respiratory complex I

Karolina Szczepanowska [1,2✉], Katharina Senft[1,2], Juliana Heidler[3], Marija Herholz[1,2], Alexandra Kukat[1,2], Michaela Nicole Höhne[4], Eduard Hofsetz[1,2], Christina Becker[1,2], Sophie Kaspar[1,2], Heiko Giese[5], Klaus Zwicker [6], Sergio Guerrero-Castillo [7,8], Linda Baumann[1,2], Johanna Kauppila[9], Anastasia Rumyantseva [1,2], Stefan Müller[1], Christian K. Frese[1], Ulrich Brandt [7,8], Jan Riemer[4], Ilka Wittig[3] & Aleksandra Trifunovic [1,2✉]

Regulation of the turnover of complex I (CI), the largest mitochondrial respiratory chain complex, remains enigmatic despite huge advancement in understanding its structure and the assembly. Here, we report that the NADH-oxidizing N-module of CI is turned over at a higher rate and largely independently of the rest of the complex by mitochondrial matrix protease ClpXP, which selectively removes and degrades damaged subunits. The observed mechanism seems to be a safeguard against the accumulation of dysfunctional CI arising from the inactivation of the N-module subunits due to attrition caused by its constant activity under physiological conditions. This CI salvage pathway maintains highly functional CI through a favorable mechanism that demands much lower energetic cost than de novo synthesis and reassembly of the entire CI. Our results also identify ClpXP activity as an unforeseen target for therapeutic interventions in the large group of mitochondrial diseases characterized by the CI instability.

[1] Cologne Excellence Cluster on Cellular Stress Responses in Aging-Associated Diseases (CECAD) and Center for Molecular Medicine (CMMC), University of Cologne, 50931 Cologne, Germany. [2] Institute for Mitochondrial Diseases and Aging, Medical Faculty, University of Cologne, 50931 Cologne, Germany. [3] Functional Proteomics, ZBC, Faculty of Medicine, Goethe University, 60590 Frankfurt am Main, Germany. [4] Department of Chemistry, Institute of Biochemistry, University of Cologne, 50674 Cologne, Germany. [5] Molecular Bioinformatics, Goethe-Universität Frankfurt am Main, 60325 Frankfurt am Main, Germany. [6] Institute of Biochemistry I, Faculty of Medicine, Goethe-University Frankfurt, 60590 Frankfurt am Main, Germany. [7] Nijmegen Centre for Mitochondrial Disorders, Radboud University Medical Center, 6525 GA Nijmegen, The Netherlands. [8] Cluster of Excellence Frankfurt "Macromolecular Complexes", Goethe-University, 60590 Frankfurt am Main, Germany. [9] Department of Mitochondrial Biology, Max Planck Institute for Biology of Aging, 50931 Cologne, Germany. ✉email: karolina.szczepanowska@uk-koeln.de; aleksandra.trifunovic@uk-koeln.de

Mitochondria are the central hub for metabolism, providing the majority of cellular ATP yield through the action of a respiratory chain coupled to the oxidative phosphorylation (OXPHOS). The oxidation of NADH by Complex I (CI, NADH:ubiquinone oxidoreductase), the largest enzyme of the respiratory chain and an essential part of supercomplexes, provides the prevailing driving force for ATP synthesis[1,2]. CI deficiency is the most common OXPHOS pathology that has been implicated in the pathogenesis of mitochondrial diseases, Parkinsonism, diabetes, cancer, and aging[3–5]. Mammalian CI consists of 45 subunits organized in a modular, L-shaped, membrane-rooted structure[6,7]. MtDNA-encoded subunits (ND1-ND6, ND4L) together with 21 supernumerary, nuclear-encoded subunits form the membrane arm (P-module) that allows proton pumping across the inner mitochondrial membrane[6,8,9]. The matrix exposed peripheral arm is composed of 17 nuclear-encoded subunits divided into two functional parts: a matrix proximal NADH-oxidizing-N-module, and a membrane adjacent ubiquinone-reducing-Q-module. Altogether, they contain a chain of eight iron-sulfur (FeS) clusters wiring the electron flow through the complex[1].

The peripheral arm, especially the flavin mononucleotide (FMN) cofactor site in the N-module, is also the primary site of CI reactive oxygen species (ROS) production, making it particularly prone to intrinsic damage[10,11]. Thiol-based redox modifications have been proposed as a ROS-counteracting, self-regulating strategy that transiently reduces electron transfer within CI to prevent cell and tissue damage[12]. Therefore, constant surveillance and replacement of the N-module should be critical to maintain CI function while preventing extensive oxidative damage and might be achieved through matrix quality control machinery.

Although the principles of stepwise CI biogenesis have been recently described in great detail[9], the regulation of its turnover, remodeling, and stability remains enigmatic. Some studies suggested that the stability and remodeling of CI might be implicated in the regulation of supercomplex formation, response to hypoxia, and in a fine-tuning of mitochondrial OXPHOS to the available carbon source[13–15]. While it was proposed that the membrane-embedded CI components are actively removed by iAAA and mAAA proteases[16,17], the machinery and mechanisms that allow selective clearance of the CI peripheral arm under physiological condition remain mostly unknown. This is surprising given that the N-module is considered to be the primary site of ROS production, and therefore most likely to suffer the potential ROS-related damage.

Here, we demonstrate that mitochondrial ClpXP protease is required for the turnover of the core part of the N-module of CI, which occurs at a higher rate and largely independently of the rest of the complex. The regular replacement of the core N-module components is required to prevent the inactivation of CI, which results in defective respiration. Moreover, it acts as a safeguard mechanism, which, when OXPHOS is stalled, prevents the electron flow through the respiratory chain, thereby limiting the production of ROS.

## Results

**N-module subcomplexes accumulate upon CLPP deficiency**. To study the CI dynamics, we analyzed protein exchange rates using pulsed Stable Isotope Labeling with Amino Acids in Cell Culture (SILAC) complexome analysis in non-dividing, differentiated C2C12 myoblasts. Overall, CI turned over rather slowly in postmitotic cells, as after 7 h, the newly synthesized proteins contributed to barely 4% of the fully assembled CI (Fig. 1a and Supplementary Data 1). Remarkably, peripheral arm subunits

presented a much higher incorporation rate than those of the membrane arm, with the N-module subunits being replaced on average six times more, and the Q-module three times more than P-module subunits (Fig. 1a and Supplementary Data 1).

In a recent screen for putative ClpXP substrates, we identified several CI peripheral arm components, including all three core subunits of N-module (NDUFV1, NDUFV2, and NDUFS1)[18]. ClpXP is multiunit proteasome-like machinery in the mitochondrial matrix composed of one or two hexamers of the AAA+ ATPase CLPX, and a tertadecameric serine protease CLPP[19]. To assess whether the CLPP protease is responsible for the rapid removal of N-module components, we analyzed the turnover rate of newly synthesized NDUFV2 using pulse-chase immunoprecipitation (Fig. 1b). Indeed, we detected a robust stabilization of NDUFV2 upon CLPP depletion, while confirming a rapid degradation of the protein in the presence of CLPP (Fig. 1b).

In the absence of CLPP in cardiomyocytes, N-module subunits accumulate in several distinct subcomplexes, the most prominent of which contains all three, core subunits: NDUFV1, NDUFV2, and NDUFS1 (Fig. 1c). In contrast, a putative CLPP substrate NDUFS2 and Q-module subunit was not detected in additional subcomplexes (Fig. 1c).

To gain a better insight into assembly intermediates that might be accumulating in the absence of CLPP, we performed high-resolution complexome profiling analysis[9]. A high, 11-fold accumulation of the core N-module subcomplex composed of NDUFV1, NDUFV2, NDUFS1, and NDUFA2 subunits confirmed our previous analysis (Fig. 1d, Supplementary Fig. 1a and Supplementary Data 2). This very robust accumulation of the N-module intermediates cannot be solely explained by augmented CI assembly because other complexes, including the MCIA complex (mitochondrial CI assembly factor complex), and the most prominent Q-module subcomplex, showed only a moderate twofold increase in the CLPP-deficient heart mitochondria. In contrast, the amount of P-module assembly intermediates remained unchanged (Fig. 1e, Supplementary Fig. 1b, and Supplementary Data 2). Furthermore, we observed a 50% decrease of fully assembled CI, while levels of CIII and CIV were much less affected (Fig. 1f, Supplementary Fig. 1a, and Supplementary Data 2). Consequently, loss of CI in CLPP-deficient cardiomyocytes led to decreased levels of supercomplexes, while the individual CIII and CIV proportionally accumulated (Supplementary Fig. 1c and Supplementary Data 2). Still, putative ClpXP substrates (NDUFV1, NDUFV2, NDUFS1, NDUFS2, NDUFAB1) showed significantly higher abundance when compared to other CI subunits (NDUFS4, NDUFS3, or NDUFB11) (Fig. 1g and Supplementary Fig. 1d).

The overall decrease in the amount of assembled CI in the heart was confirmed by the analysis of all FeS clusters in CI, as detected by the EPR (electron paramagnetic resonance) spectroscopy[20] in CLPP-deficient mitochondrial membranes (Fig. 1h). In contrast, the FeS cluster S3, inside complex II, had almost identical intensity in both samples (Supplementary Fig. 1e). In agreement with the observed loss of CI, and therefore FeS clusters, we detected significant upregulation of several enzymes involved in the synthesis and trafficking of FeS clusters (Supplementary Fig. 1f and Supplementary Data 3).

**Subunits of N and P modules follow different exchange paths**. To provide further evidence for the role of ClpXP protease in the turnover of the core N-module subunits, we inhibited CI assembly by preventing the protein synthesis inside mitochondria, which is needed to produce seven essential P-module subunits. Upon inhibition of mitochondrial protein synthesis with doxycycline or chloramphenicol, parallel to a gradual loss of

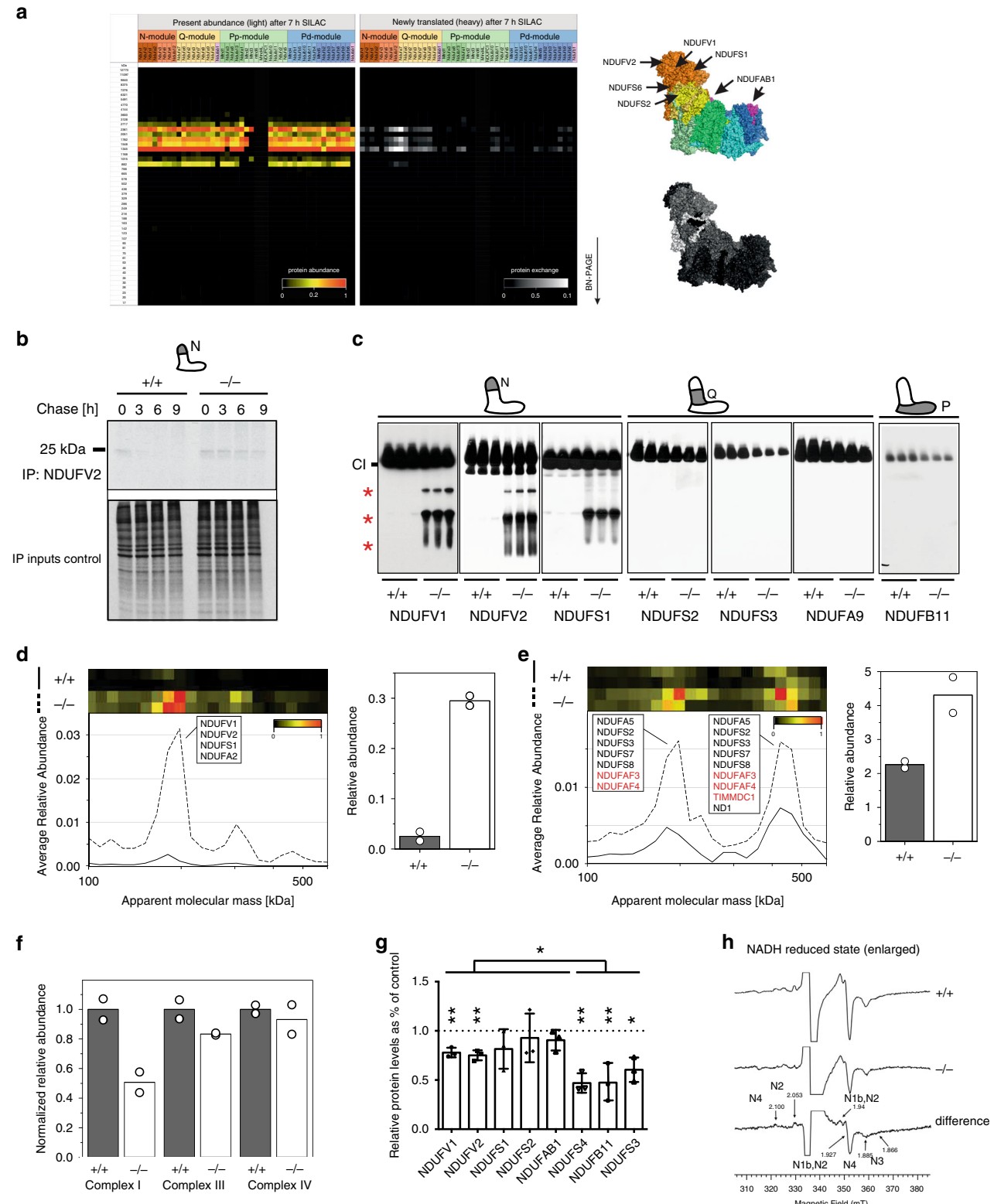

assembled CI, N-module subcomplexes strongly accumulated in CLPP-deficient mouse embryonic fibroblasts (MEFs) (Fig. 2a and Supplementary Fig. 2a). To test this further, we used the CRISPR/Cas9 approach to create cell lines lacking NDUFB11 in wild type and CLPP-deficient background. In line with previous studies, *Ndufb11* deletion resulted in a complete loss of CI by preventing its assembly[8], while the free N-module was strongly preserved by

CLPP deficiency in DKO animals (Fig. 2b and Supplementary Fig. 2b).

Accumulation of CI subcomplexes might also be a consequence of overall misbalanced protein synthesis or impaired import of newly synthesized subunits[21]. However, the analysis of global protein synthesis in CLPP-deficient cells showed mild changes, with no effect on nuclear-encoded OXPHOS subunits, including

**Fig. 1 Components of Complex I peripheral arm accumulate upon CLPP deficiency. a** Incorporation rates of CI subunits in differentiated C2C12 cells. Mitochondrial OXPHOS complexes were separated with BN-PAGE and examined by pulsed SILAC complexome analysis. Heatmap represents steady-state protein abundance (left) and assembly of newly synthesized subunits in the existing CI (right). Positions of putative ClpXP substrates (top), and visualization of exchange rates of individual CI subunits (bottom) are presented on the CI structure according to ref. [6]. **b** Turnover dynamics of NDUFV2 in wild type (+/+) and CLPP-deficient (−/−) MEFs analyzed by pulse-chase immunoprecipitation. Cells were labeled with [35]S-methionine, proteins were isolated at indicated time points after cold-chase and immunoprecipitated with anti-NDUFV2 antibody, followed by SDS-PAGE. Relevant input fractions were used as controls. (n = 3, biologically independent experiments). **c** BN-PAGE followed by western blot analysis of CI in wild type (+/+) and CLPP-deficient (−/−) hearts. Fully assembled CI and N-module containing subcomplexes (asterisks) are indicated. Antibodies used are indicated in the Figure, with putative CLPP substrates shown in bold. Individual lanes represent independent biological replicates. (n = 3, biologically independent experiments). **d–f** Migration profiles and corresponding quantifications of N-module (**d**) and Q-module (**e**) subassemblies in wild type (+/+) and CLPP-deficient (−/−) heart mitochondria. **f** Relative abundance of mitochondrial respiratory complexes in heart mitochondria. Segments of complexome profiles were derived from high-resolution supramolecular complexome profiling. Bars represent mean of n = 2 biologically independent samples. **g** Steady-state levels of individual CI subunits and assembly factors in wild type (+/+) and CLPP-deficient (−/−) heart mitochondria. VDAC was used as a loading control. CLPP substrates are shown in bold. Bars represent mean ± SD (*p < 0.05, **p < 0.01). Unpaired Student's t-test was used to determine the level of statistical difference (n = 3; biologically independent samples). **h** EPR spectra of mitochondrial membranes upon reduction with NADH in wild type (+/+) and CLPP-deficient (−/−) heart mitochondria. Signals from complex I Fe-S clusters can be identified at the low and high-field site of the predominant S3 signal originating from CII. The indicated g-values in the expanded difference spectrum mainly represent signal contributions of clusters N4, $g(z) = 2.100$ and $g(x) = 1.885$; N2, $g(z) = 2.053$; N3, $g(x) = 1.866$; several clusters are involved at $g = 1.94$ and $1.927$ but the signal results mainly from N1b $g(x,y)$ and N2 $g(x, y)$. Spectra were normalized to the same protein concentration; ten scans were accumulated for each spectrum.

CI components (Supplementary Fig. 2c and Supplementary Data 4). Next, we tested the import dynamics of newly synthesized CI subunit (NDUFV2 and NDUFS3) precursors into freshly isolated heart mitochondria and their subsequent incorporation into CI. Upon successful import into mitochondria (Fig. 2c), both proteins were promptly detected in the fully assembled CI (Fig. 2d). This result indicates the rapid exchange of N- and Q-module subunits in the pre-existing CI. However, the NDUFV2 replacement dynamics was visibly slower in CLPP-deficient mitochondria, suggesting that the protease activity might be needed for the efficient replacement of its CI substrates (Fig. 2d).

Instead, when we followed the exchange dynamics of newly synthesized, mtDNA-encoded P-module subunits, we detected several assembly intermediates and much slower integration of these subunits into fully assembled CI, but no substantial effect of CLPP deficiency (Fig. 2e). Two orders of magnitude quicker incorporation time of newly synthesized N- and Q-module subunits into CI, compared to mtDNA-encoded P-module subunits, suggests that peripheral arm proteins are steadily exchanged into pre-existing CI. In contrast, the membrane-embedded components seem to follow a different route and need de novo synthesis of whole CI (Fig. 2d, e).

**Inactive N-module is steadily exchanged on pre-existing CI.** When CLPP is depleted in cardiomyocytes, the removal of 12s rRNA chaperone ERAL1 from the small ribosomal subunit (28s) is delayed, resulting in lower levels of mitoribosome and slower mitochondrial translation[18,22] Consequently, a lack of mtDNA-encoded subunits contributes to the respiratory chain deficiency observed in CLPP-deficient hearts[18]. To distinguish between the physiological consequence of the translation defect, and the CI defect described here, we turned to CLPP-deficient cells. Although some CLPP-deficient MEFs partially recapitulate the translational defect observed in heart mitochondria, for all subsequent analyses, we choose lines that did not present this phenotype (Fig. 3a). Similarly, HeLa (and HEK293) cells lacking CLPP did not show a defect in mitochondrial translation (Supplementary Fig. 3a). Unlike in cardiomyocytes, in CLPP-deficient or CLPX-depleted MEFs and HeLa cells, we observed mostly normal levels of individual CI subunits, while putative CLPP substrates (NDUFV1 and NDUFV2) accumulated (Supplementary Fig. 3b). Remarkably, CLPP deficiency in both MEFs and HeLa cells led to a significant decrease in basal and maximal

oxygen consumption rates (OCRs). At the same time, glycolysis seems not to be affected, as no difference in the extracellular acidification rate was detected (Fig. 3b and Supplementary Fig. 3c).

We next used pulse SILAC coupled to high-resolution complexome analysis to investigate replacement rates of CI subunits. Remarkably, CLPP-deficient cells have normal steady-state levels of fully assembled CI (Fig. 3c, upper panels, and Supplementary Data 5). In these proliferating cells, the overall exchange rate of the fully assembled CI did not differ between CLPP-deficient and wild type MEFs (Fig. 3d, and Supplementary Fig. 3d and Supplementary Data 5). However, core N-module subunits were exchanged at a significantly lower rate than the rest of the subunits into fully assembled CI in the absence of CLPP (Fig. 3d, Supplementary Fig 3d, and Supplementary Data 5).

Next, we looked at the presence of N-module subunits in different subcomplexes to better understand the mechanism behind their slow exchange on the assembled CI. In wild type mitochondria, newly synthesized N-module core subunits incorporated rapidly into assembled CI and were present at very low levels as a free N-module (Fig. 3e, Supplementary Fig 3d, and Supplementary Data 5). In contrast, in CLPP-deficient mitochondria, newly synthesized subunits were exchanged very poorly at fully assembled CI and consequently mostly accumulated as free N-module (Fig. 3c, e, and Supplementary Fig 3d). These data suggest that ClpXP plays an active role in the removal of the inactive N-module from the CI, and might even be required for the assembly of N-module into CI. If this is true, in the absence of proteolytic component (CLPP), the substrate recognition part of the protease (CLPX) might become "trapped" on CI. In agreement, CLPX co-migrated with a fully assembled CI, as well as fractions containing N-module subcomplexes (Supplementary Data 6). CLPX might even act as an N-module chaperone that stabilizes its binding to CI, in the absence of CLPP. Indeed, we observed a higher accumulation of free N-module, when CLPX was depleted in CLPP-deficient cells (Fig. 4a).

To trace the origin of the OCR defect, we looked at possible structural changes of CI subunits by following cysteine modification using inverse redox shift assays. Cysteine modifications of N-module subunits have been shown to cause irreversible conformational changes that lead to CI inactivation[23]. Remarkably, all tested ClpXP substrates seem to have several cysteine residues that could be modified, in contrast to CI subunits that are not degraded by the protease (Fig. 4b, c). In wild-type

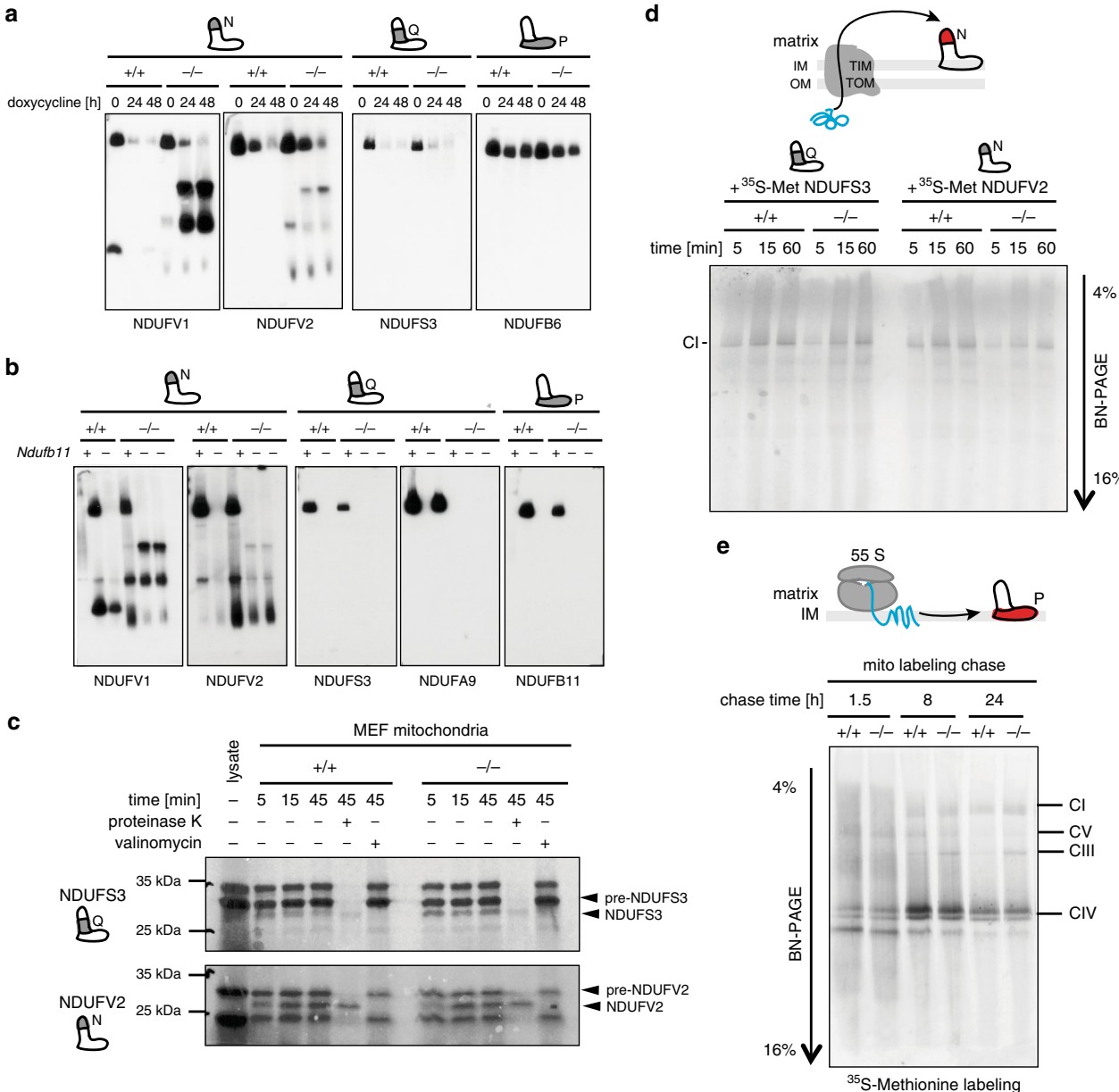

**Fig. 2 ClpXP protease regulates the turnover of N-module. a** Time-lapse BN-PAGE followed by western blot analysis of CI profiles in wild type $(+/+)$ and CLPP-deficient $(-/-)$ MEFs upon acute respiratory chain disruption via doxycycline-mediated inhibition of mitochondrial protein synthesis $(n = 3)$. **b** BN-PAGE followed by western blot analysis of CI in wild type $(+/+)$ and CLPP-deficient $(-/-)$ cells upon CRISPR/Cas9 depletion of NDUFB11 subunit $(n = 3)$. **c** Import of radiolabeled precursors (pre-) of CI subunits into wild type $(+/+)$ and CLPP-deficient $(-/-)$ mitochondria isolated from MEFs $(n = 4)$. **d** Import of radiolabelled NDUFV2 and NDUFS3 precursors and subsequent incorporation into Complex I in intact heart mitochondria from wild type $(+/+)$ and CLPP-deficient $(-/-)$ animals. After the indicated incubation times mitochondrial complexes were analyzed by BN-PAGE $(n = 4)$. **e** Pulse labeling ($^{35}$S-methionine) of mtDNA-encoded subunits in wild type $(+/+)$ and CLPP-deficient $(-/-)$ MEFs followed by chases for indicated time points. Mitochondrial complexes were analyzed by BN-PAGE $(n = 3)$. Antibodies used were raised against proteins indicated in the Figure, with putative CLPP substrates shown in bold. "$n$" represents number of biologically independent experiments.

mitochondria, these cysteine residues appeared inaccessible, likely due to disulfide bond formation and FeS cluster binding (Fig. 4b, c). Upon CLPP deficiency or CLPX depletion, a fraction of each protein emerged as modified species with cysteine residues accessible to the labeling agent (NEM) (Fig. 4b, c). Although high levels of paraquat induced presence of the same modified species, this might be secondary to the overall detrimental effect of the drug and not due to specific oxidative damage to the N-module subunits (Fig. 4b, c). In agreement with this, we neither observed similar changes upon rotenone treatment nor did depletion of

LONP1, a mitochondrial matrix protease predicted to degrade oxidized proteins, affect the accumulation of modified CI subunits (Fig. 4b, c).

In all three N-module core subunits, the majority of cysteines are bound in FeS clusters, and their higher accessibility to NEM suggested that, some FeS clusters are not present. This suggests that, either we are detecting damaged N-module subunits that lost FeS clusters and remain bound to CI, or these are newly synthesized proteins that still do not have FeS incorporated. To distinguish between these two possible scenarios, we turned to

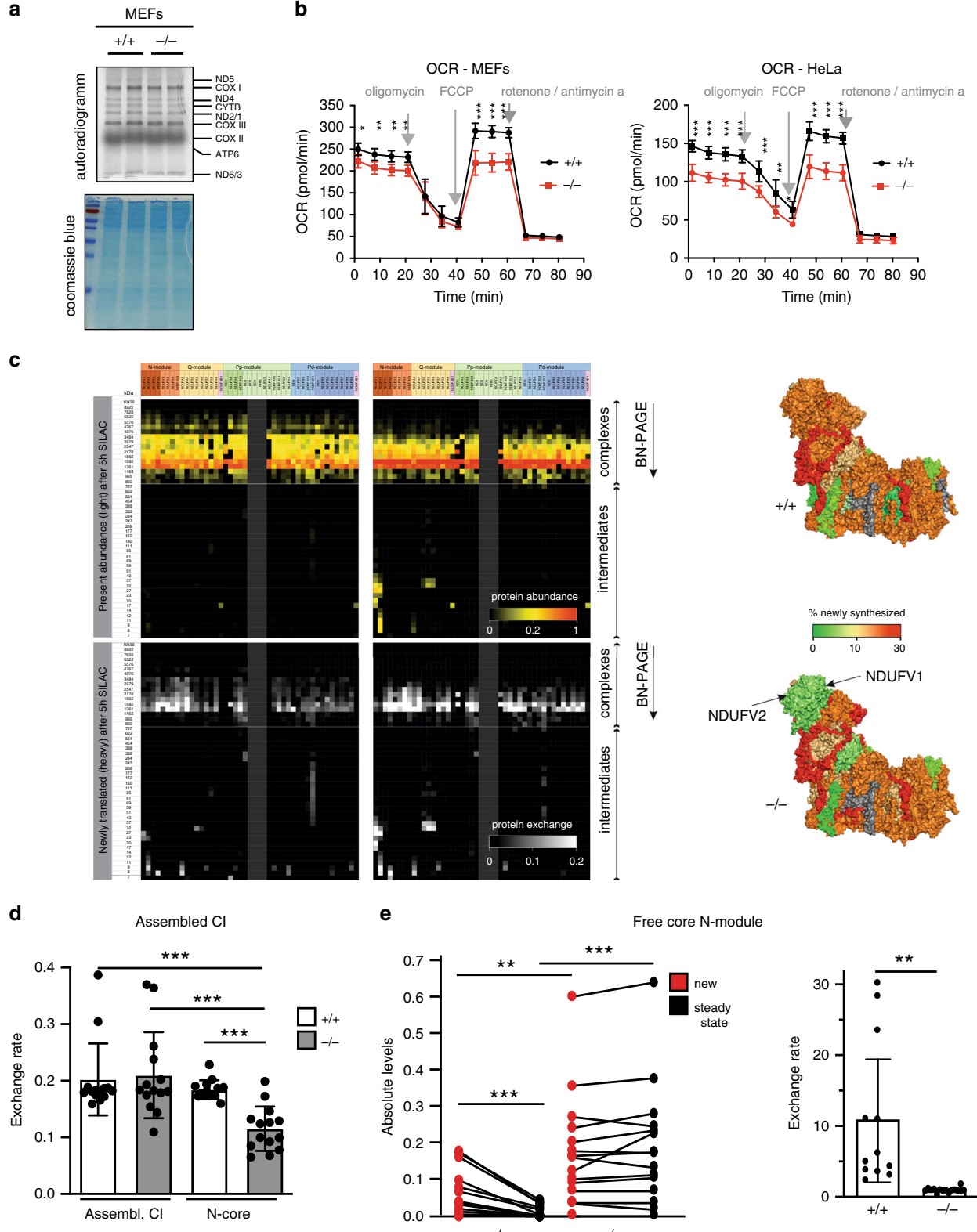

*Ndufb11/Clpp* double knockout cells (DKO). The DKO cells do not assemble CI (Figs. 2b and 4d, lower panel) and thus allowed us to analyze cysteines exclusively in free N-core subunits. The analysis of free NDUFV1 and NDUFV2 subunits in DKO cells revealed that all relevant cysteines are fully accessible to NEM, suggesting that FeS clusters are added only upon assembly of these subunits into CI (Fig. 4d). Hence, our data indicate that

CLPP deficiency does not result in the loss of FeS clusters from the fully assembled CI. Instead, the newly synthesized N-modules, which are not fully matured as they do not have incorporated FeS, accumulate as free subcomplexes.

While the loss of FeS clusters was not the primary cause of the observed CI defect, N-module inactivation seems to lead to a conformational change causing it to fall off more easily from the

**Fig. 3 In the absence of CLPP N-module is not exchanged on CI. a** De novo synthesis of mitochondrial proteins in wild type (+/+) and CLPP-deficient (−/−) MEFs followed by SDS-PAGE. Coomassie blue staining was used as a loading control. Individual lanes represent technical replicates of the wild type and CLPP-deficient MEF lines characterized by similar rates of mitochondrial translation. **b** Mitochondrial oxygen consumption rates (OCR; pmol $O_2$/min) in wild type (+/+) and CLPP-deficient (−/−) MEFs and HeLa cells. Bars represent means ± SD. (MEFs $n = 5$–6; HeLa $n = 6$; biologically independent samples). Specific inhibitors used in the analysis are indicated. **c** Turnover of the CI subunits in wild type (+/+) and CLPP-deficient (−/−) MEFs. Cells were examined using pulsed SILAC complexome analysis after separation with BN-PAGE. Heatmap (top) shows protein abundance of previously synthesized proteins (light amino acids). Heatmap (bottom) shows newly biosynthesis and assembly of subunits in existing stable CI. Right panel represents the exchange rates of individual CI subunits on a fully assembled complex visualized on the CI structures. **d, e** Exchange rates of CI subunits in wild type (+/+) and CLPP-deficient (−/−) MEFs. Cells were examined using pulsed SILAC complexome analysis after separation with BN-PAGE. Bars represent mean ± SD. (**$p < 0.01$, ***$p < 0.001$). Paired Student's $t$-test was used to determine the level of statistical difference. Detailed analysis is described in Supplementary Data 5. **d** Exchange rates of either all CI subunits (Assembl. CI) or only three (NDUFS1, NDUFV1, and NDUFV2) core N-module subunits (N-core) at the level of fully assembled CI (fractions 7628–995, $n = 14$) were calculated as ratio of newly synthesized to steady-state levels. **e** (left) Newly synthesized (new) and steady-state levels of core N-module subunits (NDUFS1, NDUFV1, and NDUFV2) at the level of free N-module subcomplexes (fractions 69–7, +/+ $n = 12$; −/− $n = 15$); (right) Exchange rates of core N-module subunits calculated as ratio of newly synthesized (red) to steady state (black) levels.

CI when mitochondria are treated with detergents or increased NaCl concentrations (Fig. 4e). This mechanic instability could be caused by N-module damage from shear stress or redox-dependent loss of flavin.

To determine whether the loss of flavin contributes to the N-module turnover, we estimated the amount of FMN in intact mitochondria and alamethicin-permeabilized mitochondrial membranes[24] of wild type and CLPP-deficient cells. This analysis showed that CLPP-deficient mitochondria indeed have lower FMN content when compared to controls (Fig. 4f). The treatment with mitoPQ, which was previously shown to stimulate the superoxide production selectively through the FMN site of CI[25], led to a severe loss of FMN only from the CLPP-deficient mitochondria (Fig. 4f). This result indicates that the CLPP-mediated CI salvage pathway can considerably compensate for the loss of FMN cofactor caused by extensive ROS production. No difference in the FMN content in NDUFB11KO and DKO mitochondria suggests that free N-modules do not contain FMN cofactor, which, like FeS clusters, might only be incorporated once N-module is assembled on CI (Fig. 4f). Alternatively, the FMN incorporated into free N-module subunits might be unstable if not rapidly transferred to CI. Nevertheless, this result must be interpreted with caution as the obtained concentrations were at the lower limit of detection.

**N-module inactivation limits oxidative stress.** To further understand the dynamics and potential triggers of N-module turnover, we first tested whether general OXPHOS impairment would lead to its loss. The treatment with potent OXPHOS inhibitors and ROS inducers, rotenone (CI) and antimycin A (CIII), led to a further accumulation of subcomplexes lacking the N-module in CLPP-deficient cells (Fig. 5a, b and Supplementary Fig. 4a–c). Dissociation of the N-module caused by rotenone and antimycin A treatment also resulted in the accumulation of CI subcomplex containing only the Q- and P-module (Fig. 5b). The same phenotype was observed in the presence of cycloheximide, a potent inhibitor of cytoplasmic translation, arguing that it cannot be the result of defective assembly (Supplementary Fig. 4d).

Interestingly, the accumulation of free N-module components and less active CI proved to be advantageous for cell survival, as CLPP-deficient cells showed better survival rates upon rotenone treatment (Fig. 5c). Nevertheless, neither accumulation of N-module subcomplexes, nor survival rates were affected by treatment with N-acetyl-L-Cysteine (NAC), a compound that increases cellular thiols (e.g., GSH) thereby decreasing overall oxidation levels (Fig. 5a–c). Similarly, prevention of cysteine reduction by the addition of dithiothreitol (DTT) could not revert the CI breakdown inflicted by rotenone or antimycin A treatment

(Supplementary Fig. 4c). Surprisingly, treatment with paraquat, a potent oxidizing agent that does not primarily affect the OXPHOS function did not result in further accumulation of free N-module (Supplementary Fig. 4d). Bypassing the $NAD^+$ shortage, commonly associated with the CI deficiency, similarly did not affect N-module stability (Fig. 5a). Furthermore, treatment with BCNU, an inhibitor of glutathione reductase, or auranofin, an inhibitor of thioredoxin reductase, both of which lead to an increased oxidation status of the cell, did not cause additional accumulation of N-module in CLPP-deficient cells (Supplementary Fig. 4e).

Collectively, these results suggested that, either only ROS produced inside the respiratory chain might be critical for the accumulation of free N-module, or that the N-module dissociation from CI is caused by conformational changes induced by OXPHOS perturbations, rather than oxidative stress per se.

To understand if OXPHOS dysfunction uncoupled from massive ROS production also leads the N-module dissociation, we treated cells with myxothiazol or potassium cyanide (KCN), (a potent CIII and CIV inhibitor, respectively) that cause a strong OXPHOS inhibition, yet induce very little to no ROS production inside mitochondria[26]. Nevertheless, both myxothiazol and KCN caused further accumulation of free N-module in CLPP-deficient MEFs, demonstrating that OXPHOS impairment without ROS production also leads to a higher turnover of N-module (Fig. 5d). Collectively, these data suggest that OXPHOS stalling rather than mitochondrial ROS production might be the primary signal for N-module dissociation and degradation.

To understand if the rapid turnover of the N-module might play a protective role in cell physiology, we looked at oxidative damage and ROS production that often results from CI dysfunction[27]. We first traced the most common permanent protein oxidation adducts in CLPP-deficient and control complexome profiles (Supplementary Fig. 5a). Although we have detected oxidative changes in several CI subunits, no significant difference has been observed between wild type and CLPP-deficient mitochondria (Supplementary Fig. 5a and Supplementary Data 7). We next analyzed reversible redox changes in the proteome of control and CLPP-deficient cells. Upon CLPP depletion, 40 proteins appeared significantly more, and 61 significantly less oxidized, suggesting that the CLPP loss and CI inactivation leads to overall lower oxidation of mitochondrial proteins (Fig. 5e, Supplementary Fig 5b, and Supplementary Data 3 and 8). Oxidized thiols were also detected in several CI subunits, but the levels did not differ from control hearts (Supplementary Data 3 and 8).

The presence of inactive CI in CLPP-deficient cells resulted in lower ROS production in both, steady-state condition and upon

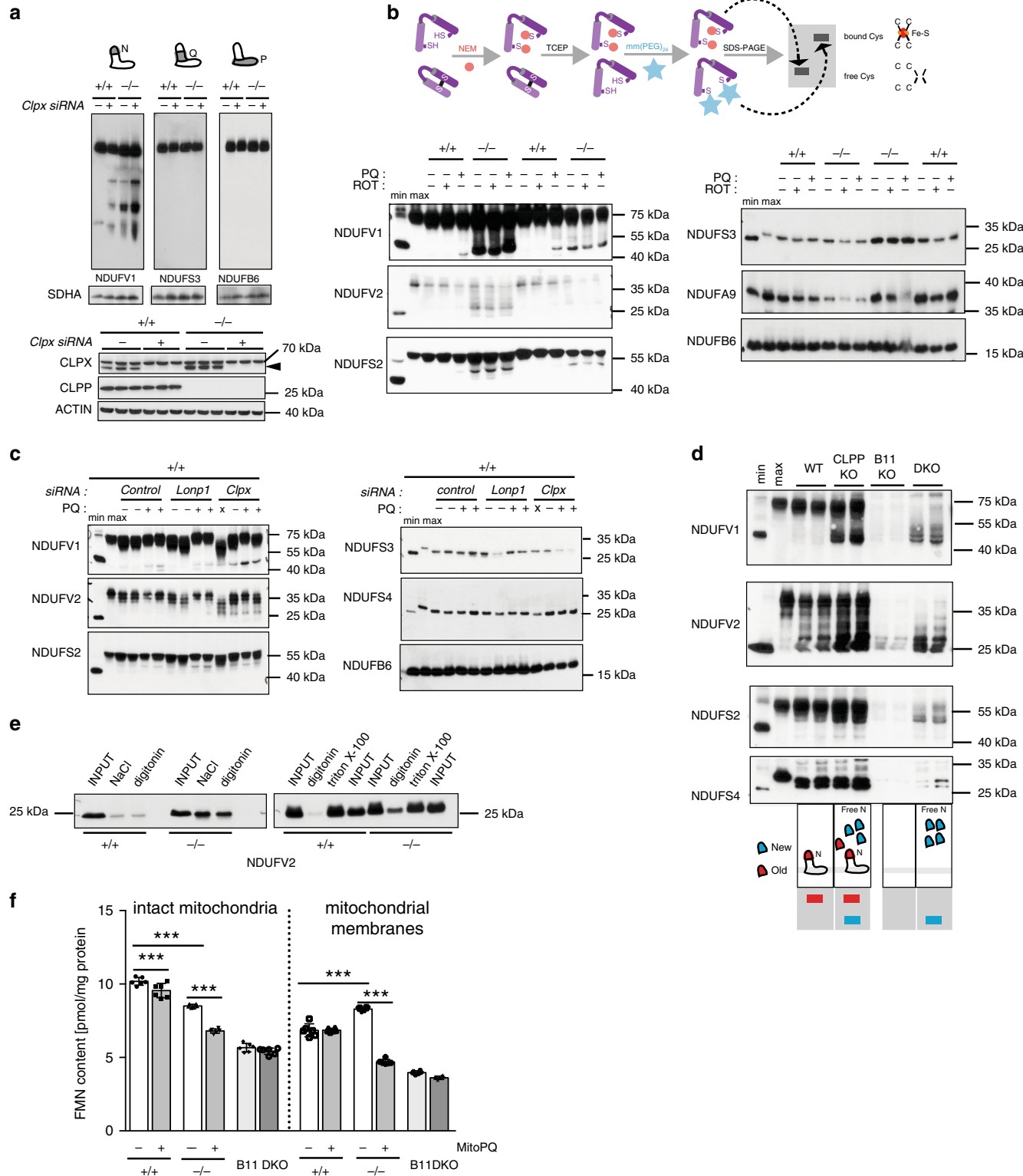

higher oxidative stress caused by treatment with increasing $H_2O_2$ concentrations (Fig. 5f and Supplementary Fig. 5c). This cannot be simply attributed to a lower CI activity, as in mtDNA mutator cells characterized by a strongly decreased CI levels, ROS levels were not different from wild type[28,29]. Instead, it appears that the inactivation of the N-module under different physiological conditions might be a regulatory process aimed to switch off the electron flow through a dysfunctional respiratory chain, thereby limiting the production of ROS.

**Loss of ClpXP ameliorates respiratory chain deficiency**. A high turnover of the N-module suggests a regulatory mechanism to keep CI fully functional under physiological conditions. We speculated that, in the case of a dysfunctional respiratory chain, this might also be a signal for increased turnover of the whole CI and could further contribute to the deleterious phenotype in mitochondrial mutants. Therefore, the stabilization of the N-module induced by the loss of CLPP might provide means for CI maintainance. To test this, we used mtDNA mutator MEFs

**Fig. 4 Accumulation of modified N-module causes respiratory chain deficiency. a** Upper panel: BN-PAGE followed by western blot analysis of CI profiles upon the siRNA-mediated *Clpx* knockdown in wild type (+/+) and CLPP-deficient (−/−) MEFs. Lower panel: western blot analysis of CLPP and CPLX levels upon siRNA-mediated knockdown of *Clpx*. (*n* = 4). **b–d** Inverse-shift analysis of CI subunits in wild type (+/+) or CLPP-deficient (−/−) MEFs investigated by western blot. The NEM-mediated stabilization of the free thiols and the subsequent mmPEG$_{24}$-modification after reductant treatment (TCEP) of previously blocked thiols allowed the mass-shift differentiation within the protein pool. (min = free thiols; max = bound thiols). **b** MEFs were pre-incubated in the absence or presence of ROS-inducing rotenone (ROT; 200 μM) or paraquat (PQ; 1 mM). Each genotype is represented by two independent clones (*n* = 3); **c** Analysis performed upon 48 h siRNA-mediated knockdown of *Lonp1* or *Clpx* (*n* = 2); **d** Upper panel: inverse-shift analysis of the individual CI subunits in wild type (WT), CLPP-deficient (CLPP KO), NDUFB11-deficient (B11 KO), and CLPP/NDUFB11 double-deficient (DKO) MEFs. Lower panel: schematic representation of the experimental outcome. **e** Native-immunoprecipitation efficiency of NDUFV2 subunit from wild type (+/+) and CLPP-deficient (−/−) heart mitochondria using various solubilization conditions: NaCl–salt extraction; digitonin–lysis in 1% digitonin (w/v); Triton-X-100–lysis in 1% Triton-X-100 (w/v) (*n* = 2). **a–e** Antibodies used were raised against proteins indicated in the Figure, with putative CLPP substrates shown in bold. "*n*" represents number of biologically independent experiments. Individual lanes represent biological replicates. **f** FMN content in intact mitochondria and permeabilized mitochondrial membranes from wild type (WT), CLPP-deficient (CLPP KO), NDUFB11-deficient (B11 KO), and CLPP/NDUFB11 double-deficient (DKO) MEFs. Cells were incubated for 16 h in the presence or absence of mitoPQ (5 μM) prior to mitochondria isolation. Mitochondria from different experiments (*n* = 3–6) were pooled and further preceded as technical replicates (*n* = 6). Flavin content was determined with fluorescence record (at excitation/emission 470/525 nm), and calculated according to calibration with FMN standards. Bars represent mean ± SD (***$p < 0.001$). Two-way ANOVA followed by Tukey's multiple comparisons post-test was used to determine the level of statistical difference.

characterized by high levels of mtDNA mutations leading to instability of OXPHOS complexes and significantly slower cell proliferation[29]. CLPX depletion in mtDNA mutator cells led to a robust stabilization of individual peripheral arm subunits with a prominent effect on some non-substrate components (e.g., NDUFS3) (Supplementary Fig. 6a). Consequently, loss of ClpXP resulted in the stabilization of fully assembled CI and improved cell proliferation capacity in mtDNA mutator MEFs (Fig. 6a, b).

The majority of CI firmly associates with CIII$_2$ or CIII$_2$+CIV to form different supercomplexes. In the absence of CYTB, CIII is not made, leading to CI instability and rapid degradation[30]. Similarly, CI becomes highly unstable in cells that do not assemble CIV due to deficiency of COX10, one of the early CIV assembly factors[31]. To test if the loss of CLPP might stabilize CI in CIII and CIV mutants, we depleted ClpXP in CYTB-deficient mouse L929 cells[30] and COX10-deficient skin fibroblasts[31] (extended date Fig. 6b). Our results show that loss of ClpXP increases levels of fully assembled CI in both mutant cell lines, without having a significant effect on the stability of CIII or CIV (respectively) (Fig. 6c, and Supplementary Fig. 6c, d). However, diminished cell proliferation capacity was not changed as the underlying defect in CIII or CIV assembly was not corrected (Supplementary Fig. 6c, d).

To test if CLPP depletion could rescue the CI instability in vivo, we analyzed *gas-1(fc21)* (NDUFS2 homolog in *Caenorhabditis elegans*) mutant worms. Mitochondrial dysfunction in worms leads to strong delay in the development into a fully functional adult. However, knockdown of *clpp-1* levels in *gas-1(fc21)* mutants resulted in a quicker development, accompanied by a mild recovery of CI levels (Fig. 6d, e). Furthermore, a decreased number of progeny[32], another characteristic of most mitochondrial mutants, was partially rescued by CLPP-1 depletion (Fig. 6d). As the mutation in *gas-1(fc21)* animals directly affects CI function, a mild increase in CI levels likely does not result in improved CI activity. Unchanged oxygen consumption rates and ATP levels in *gas-1* mutants upon CLPP depletion further confirmed this hypothesis (Supplementary Fig. 6e).

## Discussion

Collectively, our data provide strong evidence for a comprehensive model of CI maintenance, acting through surveillance and replacement of expended core part of N-module by ClpXP protease. To regenerate stalled CI, ClpXP can recognize, disassemble, and rapidly degrade impaired core N-module proteins, allowing for an effective exchange with fresh N-modules components in the pre-existing CI (Fig. 6f). When ClpXP is not present, the accumulation of inactive CI subunits impairs electron flow, thereby causing OXPHOS deficiency, but also limiting deleterious ROS production (Fig. 6f). In the heart, a tissue highly dependent on respiratory chain activity, N-module inactivation induces complete CI breakdown resulting in markedly reduced CI levels and activity (Fig. 6f). We provide evidence that complete CI breakdown is initiated by the N-module dissociation as we showed that block of ClpXP-mediated N-module degradation stabilizes CI in OXPHOS-deficient models, characterized by the overall CI instability (Supplementary Fig. 6e).

The exchange of the entire N-module or its subunits in the pre-existing complex appears as a preferred and economical way of complex I maintenance that is highly dependent on ClpXP. In contrast, the integration of the mtDNA-encoded P-module sub-units requires de novo synthesis of the whole CI.

ClpXP deficiency in proliferating cells does not lead to a decrease in fully assembled CI, contrary to heart mitochondria that are characterized by an additional defect in mitoribosomal assembly, leading to 25% lower overall synthesis of mtDNA-encoded subunits[18,22]. Nevertheless, the observed 50% decrease in CI levels cannot be attributed solely to the translation defect because this would have but did not result in a corresponding CIII and CIV decrease, as was observed in other models directly affecting mitochondrial protein synthesis and mitoribosome assembly[33–35]. Therefore, we propose that in heart, and possibly other postmitotic tissues, while N-modules cannot be replaced in the absence of CLPP, accumulation of damaged CI increases the rate of complete CI degradation by the quality control system responsible for its macro-degradation. Since the overall de novo assembly of CI is unchanged, this results in a lower steady-state level of fully assembled CI in CLPP-deficient hearts. In proliferating cells, however, much higher de novo assembly rates can be expected to largely override the effect of accumulation and degradation of inactive CI.

The core N-module components and putative CLPP substrates (NDUFS1, NDUFV1, and NDUFV2) house five FeS centers and FMN, forming the principal entry gate of electron transfer and a primary site of ROS production within CI[36]. Surprisingly, very little is known about the insertion of the FeS clusters or FMN moiety into CI subunits. Recent findings indicate that Q-module FeS cluster N4 of NDUFS8 homolog in aerobic yeast *Yarrowia lipolytica* is inserted into CI at a very late assembly step[37]. The core N-module subunits with accessible cysteines detected in our shift assays seem to originate from nascent core N-modules that accumulate as free subcomplexes upon CLPP deficiency and have accessible SH groups because the FeS clusters are not yet inserted.

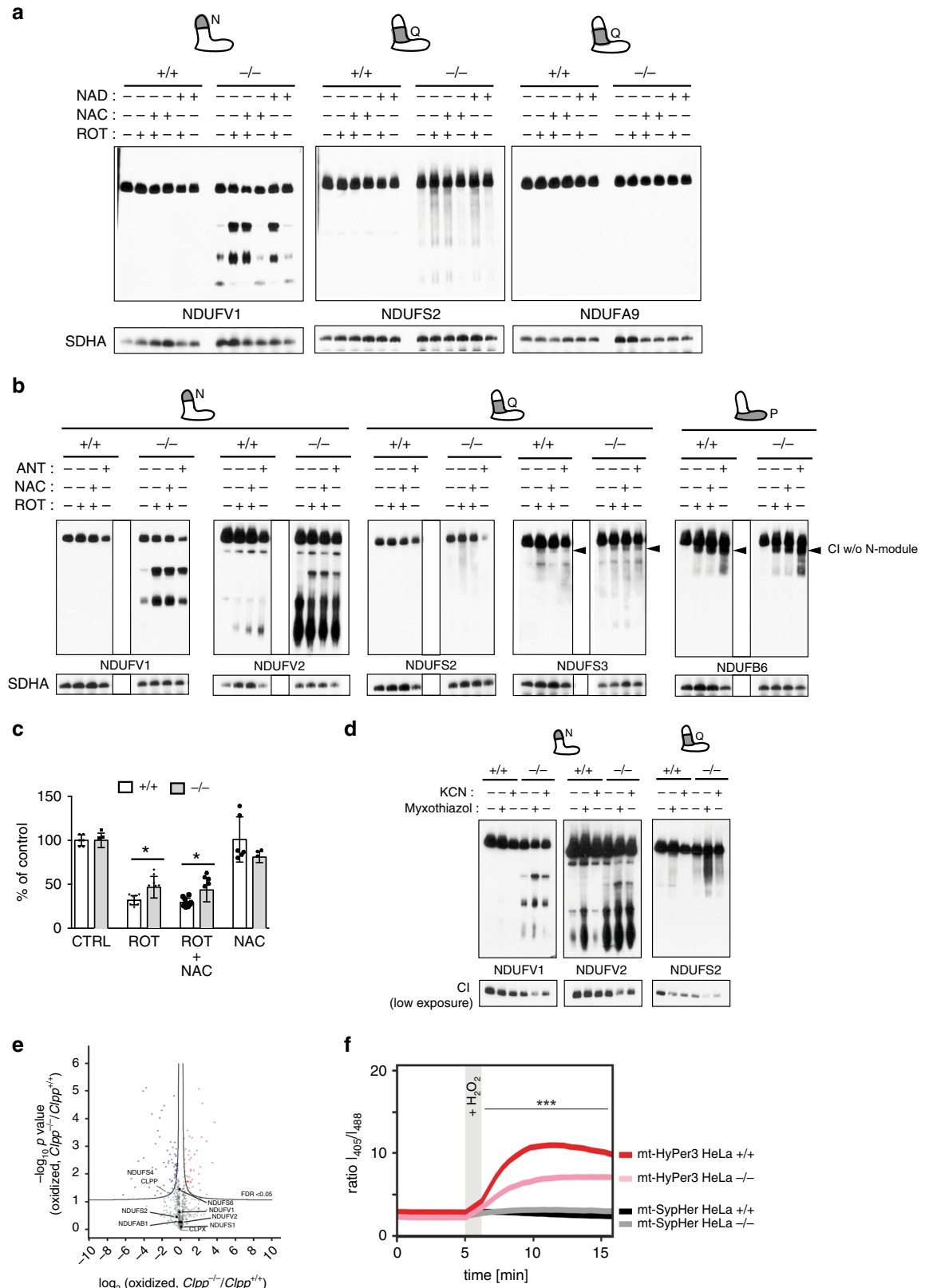

Similarly, our results suggest that the free N-module might not have FMN. Flavin moiety might be added later, or is incorporated, but unstable when N-module is not assembled directly into CI. However, this requires further confirmations as these measurements (NDUFB11 KO and DKO samples in particular) were

very challenging, and their results were at the edge of the detection limit.

The precisely coordinated, late insertion of FeS and possibly FMN into CI might be a protective mechanism that prevents newly synthesized but yet unincorporated subunits from

**Fig. 5 The N-module turnover protects from increased oxidative stress. a, b** BN-PAGE followed by western blot analysis of CI in wild type (+/+) and CLPP-deficient (−/−) MEFs treated with rotenone (ROT; 200 μM), N-acetylcysteine (NAC; 2 mM) and β-nicotinamide adenine dinucleotide hydrate (NAD; 2 mM) or antimycin A (25 μM). (n = 3, biologically independent experiments). **c** Survival of wild type (+/+) and CLPP-deficient (−/−) MEF cells upon 16 h treatment with rotenone (ROT; 200 μM) ± NAC (2 mM). Values are calculated as a percentage of the control (CTRL) cells. Bars represent mean ± SD. (*$p < 0.05$). Two-way ANOVA followed by Sidak's multiple comparison post-test was used to determine the level of statistical difference. (CTRL & NAC $n = 6$; ROT & ROT + NAC $n = 10$, biologically independent samples). **d** BN-PAGE followed by western blot analysis of CI profiles in wild type (+/+) and CLPP-deficient (−/−) MEFs treated with potassium cyanide (KCN; 1 mM) and myxothiazol (1 μM) for 16 h. (n = 2, biologically independent experiments). **a**, **b**, **d** Antibodies were raised against proteins indicated in the panels, with putative CLPP substrates shown in bold. CI depleted of N-module are indicated with the arrowheads. **e** Quantification of oxidized proteins in wild type and CLPP-deficient heart mitochondria by mass spectrometry. Values were normalized to mean protein abundance. Data are shown in volcano plots using permutation-based FDR calculation (FDR < 0.05), (n = 3, biologically independent experiments). Red color represents significantly more oxidized proteins in Clpp−/− mitochondria, and blue in Clpp+/+ samples. CLPP targets are indicated in bold. **f** Ratiometric imaging for response to hydrogen peroxide of Hela wild type and CLPP-deficient cells. Cells containing the matrix-targeted $H_2O_2$ sensor HyPer3 or as pH-control matrix-targeted SypHer sensors were analyzed for their response towards 45 μM hydrogen peroxide. The SypHer sensor was not deflected during the experiment indicating that the HyPer3 response was solely due to changes in hydrogen peroxide levels. Results were from two independent experiments (n[HeLa, HyPer3] = 141 cells, n[ClpP KO, HyPer3] = 70 cells, n[HeLa, SypHer] = 89 cells, n[ClpP KO, SypHer] = 75 cells. Shapiro–Wilk test to test for normal distribution followed by Mann–Whitney-U-test was used to determine the level of statistical difference (***$p < 0.001$).

unwanted and potentially detrimental redox activity. The N-module on assembled CI needs to be replaced at a high rate, not because it loses FeS clusters, but because it suffers either redox-dependent loss of flavin[38] or shear stress and is therefore mechanically rather unstable. The former was supported by our results showing that CLPP-deficient mitochondria are much more sensitive to redox-mediated damage of FMN. Mechanistic instability might also play a role in this process as loss of CLPX chaperone or high salt concentrations cause increased detachment of the N-module from CI.

Increased ROS levels were shown to lead to S-glutathionylation of NDUFS1 and NDUFV1 subunits, thus resulting in a general reduction of CI activity important during physiological stress scenarios like ischemia-reperfusion[23,39,40]. Structural changes seem to accompany glutathionylation, as its reversal fails to restore the CI activity. This type of CI modification seems to require replacement of inactive subunits[23], possibly mediated by selective proteolysis as described here. Supporting this notion, we show that N-module accumulation associated with CLPP deficiency is significantly boosted in conditions that cause OXPHOS damage accompanied by increased ROS production. This could neither be prevented by antioxidants nor augmented by attenuating the antioxidant machinery, suggesting that structural alterations rather than ROS production provide N-module degrons for selective ClpXP-mediated breakdown. In agreement, this mechanism seems to be also induced in conditions that cause OXPHOS stalling without an increase in ROS production.

The whole mechanism of N-module turnover is reminiscent of FtsH protease-mediated degradation of the photosystem II (PSII) reaction center protein D1[41–43]. Light energy required for the initial step of photosynthesis causes inevitable damage to the PSII complex and D1 protein in particular[43]. To maintain healthy photosynthetic activity and prevent increased oxidative stress, photosynthetic organisms have developed a sophisticated system that relies on regulated proteolysis and replacement of the D1 subunit through a process called the "PSII repair cycle"[42,43]. Although the replacement rate of D1 might be much quicker than that of the core N-module CI subunits, the basic principle is the same—centers of most extensive redox activity, in both mitochondria and chloroplasts, are more often damaged by "wear and tear", and need to be replaced more frequently to keep these essential enzymes working correctly.

Better redox fitness and lower ROS production might partly explain the beneficial effect of CLPP depletion in different mitochondrial mutants characterized by dysfunctional OXPHOS. Nonetheless, loss of CLPP also led to an evident stabilization of,

not only core N-module subunits but also the entire CI in these mutants. ClpXP is not a sole protease that degrades CI, as it cannot pull out membrane-embedded proteins, which instead seem to be degraded by i-AAA and m-AAA protease of the inner mitochondrial membrane[16,44,45]. Still, our results suggest that CI disassembly that remains an entirely unspecified process could be initiated by the ClpXP-mediated removal of the N-module. Therefore, diminishing the N-module turnover could delay and slow down the disassembly of the whole CI, resulting in an improved phenotype. However, the beneficial effect arising from the loss of ClpXP in mitochondrial mutants does not necessarily originate from the restored CI activity. This might be especially true for mutants with a primary defect in CI, like in the case of gas-1 mutant. Indeed, we showed that despite higher CI levels, no changes in overall respiration and ATP production could be detected. Thus, we propose that the beneficial effect, in this case, might be the result of the stabilization of CI-containing super-complexes leading to improved cristae morphology and, therefore, a favorable physiological outcome. Alternatively, the presence of a functional free N-module might provide means to maintain the $NAD^+/NADH$ ratio when CI is not functioning correctly.

In conclusion, we discovered a CI salvage pathway in mitochondria based on proteolytic control that allows a separate turnover of distinct N-module parts to maintain highly functional CI and healthy mitochondria. Mitochondrial ClpXP protease is central to this process as it has a role in surveillance and effective removal and degradation of expended N-module subunits. Not only is this a faster way to maintain functional CI, but it is also energetically favorable as it demands much lower expenditure than de novo synthesis and reassembly of the entire CI. Our results also highlight an unforeseen possibility for the exploration of a therapeutic intervention targeting ClpXP activity in the large group of mitochondrial diseases characterized by CI instability.

## Methods

**Maintenance of cells.** If not indicated differently, MEF, mouse skin fibroblasts, cybrids, and HEK293T cells were grown in Dulbecco's modified Eagle medium (DMEM; Gibco) supplemented with glucose (4.5 g/l), sodium pyruvate (100 μg/ml), glutamine (4 mM), fetal bovine serum (FBS) (10%), uridine (50 μg/ml) and penicillin/streptomycin. HeLa cells were grown in MEM supplemented with FBS (10%) and penicillin/streptomycin. All cell lines were routinely tested for Mycoplasm infection. The list of cell lines and information on their origin could be found in Table 1.

For drug treatments, cells were grown to 70–80% confluency in complete DMEM medium supplemented with glucose (4.5 g/l), sodium pyruvate (100 μg/ml), glutamine (4 mM), FBS (10%), and penicillin/streptomycin, and exposed to: rotenone (Sigma), antimycin A (Sigma), N-acetyl-cysteine (NAC; Sigma),

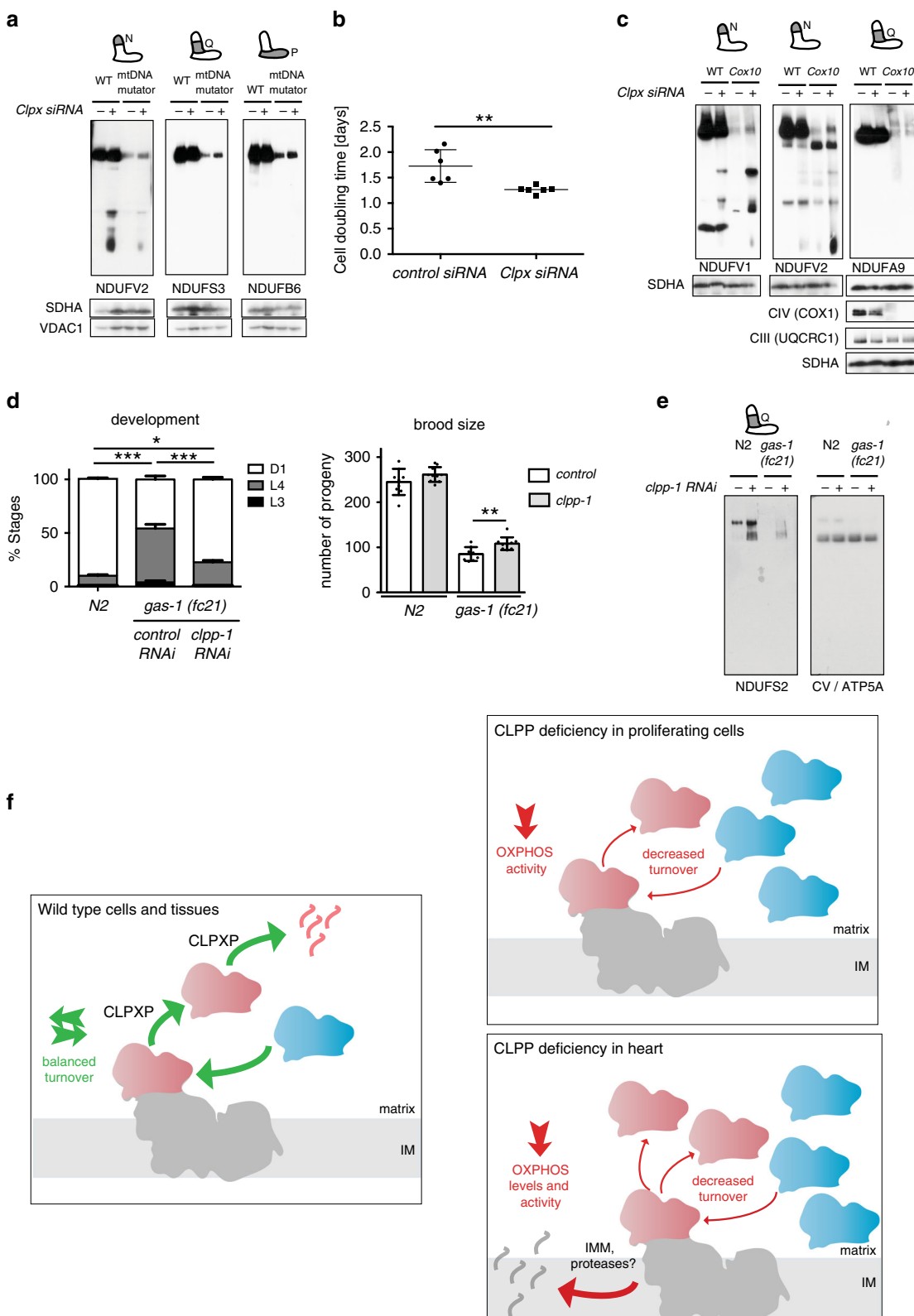

Nicotinamide adenine dinucleotide (NAD, Roche), dithiothreitol (DTT; Sigma), Carmustine (bis-chloroethylnitrosourea (BCNU, Sigma), auranofin (Sigma), myxothiazol (Sigma), potassium cyanide (Sigma), cycloheximide (Sigma), mitoparaquat (mitoPQ, Abcam), or respective vehicles at indicated concentrations and times.

To determine the cell survival, the numbers of cells were estimated following the drug treatments (or respective mock controls) using the Countees Automatic Cell

Counter combined with trypan blue staining. To determine the cell doubling rates, the equal numbers of cells were transfected with respective small-interfering RNA (siRNA) for 48 h, followed by cell number determination at the endpoint using the Countees Automatic Cell Counter combined with trypan blue staining.

**Development of new cell lines**. A 20-bp guide sequence targeting DNA within the first exon of *Ndufb11* was selected from a published database of predicted

**Fig. 6 Loss of CLPXP ameliorates respiratory chain deficiency caused by OXPHOS complex instability. a** BN-PAGE followed by western blot analysis of CI in wild type (WT) and *PolgA^mut^/PolgA^mut^* (mtDNA mutator) MEFs upon the 48 h siRNA-mediated *Clpx* knockdown. SDHA and VDAC were used as a loading control (*n* = 2, biologically independent experiments). **b** Proliferation capacity of mtDNA mutator MEFs upon siRNA-mediated *Clpx* knockdown (*Clpx siRNA*) in comparison to control (*control siRNA*) 48 h after transfection. Bars represent mean ± SD. (\*\**p* < 0.01). Unpaired Student's *t*-test was used to determin the level of statistical difference (*n* = 6, biologically independent samples). **c** BN-PAGE followed by western blot analysis of CI in wild type (WT) and CIV-deficient COX10 knockout (*Cox10*) fibroblasts upon the 48 h siRNA-mediated *Clpx* knockdown. Bottom panels show the abundance of fully assembled Complex II, III, and IV in relevant samples (*n* = 2, biologically independent experiments). **d** CLPP downregulation ameliorates the detrimental phenotypes associated with CI deficiency in roundworm *C. elegans*. Wild type (N2) and *gas-1(fc21)* mutant worms were exposed to the *Clpp*-targeting RNAi. The developmental progress was assayed as a proportion of worms that reached a particular stage four days after hatching (left panel). The brood size was estimated as a number of living progeny per adult worm. Wild-type worms were used as control. Bars represent mean ± SD (\**p* < 0.05, \*\**p* < 0.01, \*\*\**p* < 0.001). For developmental assay, statistical analysis was performed using Fisher Exact Test (*n* = 4, independent biological samples with 100 animals per condition). For brood size assay, data was compared using paired Student's *t*-test. Larval stage 3 (L3), larval stage 4 (L4), adulthood (D1) (*n* = 10, animals per condition). **e** BN-PAGE followed by western blot analysis of CI profiles in a wild type (N2) and CI-deficient (*gas-1*) worms upon the RNAi-mediated downregulation of CLPP protease. **a**–**e** Antibodies used were raised against proteins indicated in the Figure, with putative CLPP substrates shown in bold. SDHA, VDAC1, and ATP5A were used as a loading control. **f** A comprehensive model for the ClpXP-mediated CI surveillance in wild type condition, as well as in proliferating or postmitotic (heart) cells.

---

| Table 1 Cell lines. | | |
|---|---|---|
| **Name** | **Source** | **ID** |
| Human: HeLa | ATCC | CCL-2™ |
| Human: HeLa *Clpp*−/− (CRISPR/Cas9) | This paper | N/A |
| Human: HEK293T | ATCC | CRL-3216 |
| Human: HEK293T *Clpp*−/− (CRISPR/Cas9) | This paper | N/A |
| Mouse: MEFs immortalized | 18 | N/A |
| Mouse: MEFs CLPP−/− immortalized | 18 | N/A |
| Mouse: MEFs NDUFB11−/− (CRISPR/Cas9) | This paper | N/A |
| Mouse: MEFs CLPP−/−; NDUFB11−/− (CRISPR/Cas9) | This paper | N/A |
| Mouse: MEF mtDNA mutator (*Polg^D257A/D257A^*), immortalized + wt control | 28 | N/A |
| Mouse: MT-CYB-deficient mouse cybrid L929 cells + wt control | 30 | N/A |
| Mouse: COX10-deficient skin fibroblasts, immortalized + wt control | 31 | N/A |

high-specificity protospacer adjacent motif (PAM) target sites in the mouse exome (see Table 2. Primers and Oligonucleotides). Two complementary oligos containing the *Ndufb11* guide sequence and BbsI ligation adapters were synthesized, annealed and ligated into BbsI-digested pX458 vector. Wild type and *Clpp*-knockout MEFs were transfected with the *Ndufb11* single-guide (sg) RNA plasmid and Lipofectamine 2000 according to manufacturer's instructions. Single cells were seeded in 96-well plates and grown until clones appeared. Proteins from individual clones were isolated, fractionated on sodium dodecyl sulfate–polyacrylamide gel electrophoresis (SDS-PAGE) gels and screened by immunoblotting with a NDUFB11 antibody.

CLPP-deficient HEK293T and HeLa cells: for Crisp Cas9-mediated knockout of *Clpp* in HEK293T cells several strategies were used. For the classical Cas9-mediated double strand-break design the 20 bp guides binding in human *Clpp* exon 1 were cloned into plasmid pX330. For a paired nickase Cas9n approach, the 20 bp guide pairs binding in human *Clpp* exon 1 were cloned into pX335. For HeLa cells, we used the paired nickase Cas9n approach with Nick pair #7. HEK293T cells were transfected with Lipofectamine 2000 (Invitrogen) according to manufactures instructions either with the constructs pX330-*mClpp*−5 or pX330-*mClpp*−20 alone or by a 1:1 mixture of each the Nickase pairs #5 or #7 in plasmids pX335. HeLa cells were transfected with Nucleofector™ electroporation device (Lonza Group Ltd, Basel, Switzerland). Single cells were seeded in 96-well plates and grown until clones appeared. Proteins from individual clones were isolated, fractionated on SDS-PAGE gels and screened by immunoblotting with CLPP antibody. Following oligonucleotides have been used: Human *Clpp* 20-bp quide sequence #5 and #20; and Human *Clpp* Cas9n Nick pair #5: and #7 (see Table 2. Primers and Oligonucleotides).

**Mouse lines maintenance**. CLPP-deficient (−/−) mice were obtained from intercrosses of *Clpp*^+/−^ heterozygous mice lacking exons 3–5[18]. *Clpp* gene targeting was carried out on a C57BL/6NTac genetic background[18]. For all experiments, the 18–24-week-old animals of both sexes were used. All experiments were approved and permitted by the Animal Ethics Committee of North-Rhein Westphalia (Landesamt für Natur, Umwelt und Verbraucherschutz Nordrhein-Westfalen;

LANUV) following the German and European Union regulations. All animal work was performed in accordance with recommendations and guidelines of the Federation of European Laboratory Animal Science Associations (FELASA).

**C. elegans strains and assays**. Strains were cultured on OP50 *E. coli*-seeded NGM plates, according to standard protocols unless otherwise stated. To obtain *clpp-1* RNAi vector, cDNA was amplified using specific primers, cloned into an empty L4440 vector using *Kpn*I and *Not*I site and confirmed by sequencing. As a control, the empty L4440 vector was used[46].

All clones were transformed into the *E. coli* HT115 (DE3) strain. The overnight culture grown in Luria broth media was grown to OD$_{595}$ = 0.5, and then IPTG was added to a concentration of 1 mM. The bacteria were then induced for 3 h at 37 °C, shaking and seeded on NGM plates containing 100 μg/ml ampicillin, 50 μg/ml tetracycline and 1 mM IPTG. Worms were treated with RNAi from hatching and phenotype was observed as indicated.

To assay the brood size, single worms were transferred on individual plates at the L4 larval stage and allowed to lay eggs. The worms were transferred to a fresh plate every day until they stopped laying eggs. The total amount of hatched progeny was scored and plotted as total brood size. To assay developmental times, worms were synchronized using alkaline hypochlorite solution treatment, and eggs were seeded either on the empty L4440 or *clpp-1* containing plates. Seventy-two hours post bleaching, when wild-type N2 worms reached adulthood, worms were assayed for developmental stages.

For the analysis of ATP levels, worms were collected from a 10 cm plate and washed five times in M9 buffer. After the last wash, worms were resuspended in 50 μl of M9 buffer and frozen in liquid nitrogen until the experiment day. On the day of the experiment, frozen pellet was incubated on 95 °C for 15 min, and transferred immediately to ice for additional 5 min. The lysate was then centrifuged on 16,000 × *g* for 5 min with cooling. Supernatant was transferred to the fresh tubes and diluted three times. ATP levels were determined using ATP Bioluminescence Assay Kit HS II (Roche). ATP concentration was normalized to the total protein levels determined by Bradford assay.

Oxygen consumption rates were measured using an Oroboros Oxygraph 2k (Oroboros Instruments GmbH). Three-hundred animals, on the first or the fifth day of adulthood, were used for each measurement. Each measurement was performed at 20 °C and repeated at least three times. Data was analyzed using DatLab4 software (Version 4.3).

**Isolation of mitochondria**. Hearts were homogenized with a rotating Teflon potter (Potter S, Sartorious; 20 strokes, 1000 rpm) in a buffer containing 100 mM sucrose, 50 mM KCl, 1 mM EDTA, 20 mM TES, 0.2% fatty acid-free bovine serum albumin (BSA), subtilisin (1 mg per 1 g of tissue), pH 7.6, followed by differential centrifugation at 800 × *g*, and 8500 × *g*, 10 min, 4 °C.

Cultured cells were resuspended in 20 mM HEPES pH 7.6, 220 mM mannitol, 70 mM sucrose, 1 mM EDTA, 0.2% fatty acid-free BSA, and homogenized with a rotating Teflon potter (40 strokes, 1200 rpm) followed by differential centrifugation at 850 × *g*, and 8500 × *g*, 10 min, 4 °C.

Worms were collected with M9 medium, washed out from bacteria extensively, and homogenized in 220 mM mannitol, 70 mM sucrose, 10 mM Tris, 2 mM EDTA, pH 7.4 with a rotating Teflon potter (20 strokes, 1000 rpm). Homogenates were incubated with subtilisin A (10 mg/g of wet worm pellet) and incubated at 28 °C for 20 min, followed by centrifugation at 10,000 × *g* at 4 °C for 10 min. Pellet was resuspended in 220 mM mannitol, 70 mM sucrose, 10 mM Tris, 2 mM EDTA, pH 7.4 supplemented with 0.4% BSA. Worm debris was removed by centrifugation at 850 × *g* for 10 min at 4 °C, and mitochondria were pelleted by 10 min centrifugation at 10,000 × *g* at 4 °C. Mitochondria were washed with BSA-free buffer, protein concentrations were determined with Bradford reagent, and subjected to blue

**Table 2 Primers and oligonucleotides.**

| Name | ID | 5′ — ............... —3′ |
|---|---|---|
| Mouse Clpp genotyping | F | TGTGCATTCTTACCATAG TCTGC |
| | Rwt | CCCAGACATGATTCCTAGC AC |
| | RKO | CCCA GACATGATTCCTAGCAC |
| Mouse Ndufb11 guide | | TGTAATCGCCCCATCCGGTG |
| Human Clpp guide | #5 | GGTACCTGCATGACGCCACC |
| | #20 | GCAGCGGTGCCTGCACGCGA |
| Human Clpp Cas9n Nick | #5 as #5 a | GGCCCCCCCTACCAATATTC CATGCAGGTACCCCGCGCTG |
| | #7 as #7 s | TTCTGGAGTGTCCGCTGCGG GCAGCGGTGCCTGCACGCGA |
| Mouse ClpX siRNA | #1 | GCUUCGCUGUCCUAAAUGU |
| | #2 | GAUUGGCCCUGGAAAGAAA |
| Mouse Lonp1 siRNA | | CCACUCCUCUGAGUUCAAUU |
| C. elegans: clpp-1 RNAi | S | GGGGTACCATGTCTGCTTCTGTTCAATCACGCGTT |
| | A | GGTGCGGCCGCATCTGATGGCATTGATCCGTT |

**Table 3 Antibodies.**

| Antibody | Company | Cat. No. | Dilution |
|---|---|---|---|
| Rabbit polyclonal anti-CALNEXIN | Calbiochem | 208880 | 1: 1000[a] |
| Mouse monoclonal anti-HSC70 | Santa Cruz | sc7298 | 1:1000[a] |
| Mouse monoclonal anti-b-actin | Sigma | A5441 | 1:1000[a] |
| Rabbit polyclonal anti-VDAC | Cell Signalling | 4661 | 1:1000[a,b] |
| Rabbit polyclonal anti-AFG3L2 | Elena Rugarli, Univ. Cologne | N/A | 1:1000[a] |
| Mouse monoclonal anti-LONP1 | Abcam | ab82591, | 1:1000[a] |
| Rabbit polyclonal anti-CLPP | Sigma | HPA040262 | 1:1000[a] |
| Mouse monoclonal anti-ACO2 [6F12BD9] | Abcam | ab110321 | 1:1000[a] |
| Mouse monoclonal anti-ATP5A [15H4C4] | Abcam | ab14748 | 1:1000[a] 1:2500[b] |
| Mouse monoclonal anti-UQCRFS1/RISP[5A5] | Abcam | ab14746 | 1:1000[a] |
| Mouse monoclonal anti-UQCRC1 | Mol. Probes | 459140 | 1:1000[a] 1:2500[b] |
| Mouse monoclonal anti-COX4L1 | Mol. Probes | A21348 | 1:1000[a] |
| Mouse monoclonal anti-MT-CO1 | Mol. Probes | 459600 | 1:1000[a] 1:2500[b] |
| Mouse monoclonal anti-SDHA | Mol. Probes | 459200 | 1:5000[a,b] |
| Mouse monoclonal anti-NDUFB6 | Invitrogen | A21359 | 1:5000[a,b] |
| Rabbit polyclonal anti-NDUFAB1 | Abcam | ab96230 | 1:1000[a,b] |
| Mouse monoclonal anti-NDUFS4 [2C7CD4AG3] | Abcam | ab87399 | 1:1000[a,b] |
| Rabbit monoclonal anti-NDUFAF2/Mimitin | Abcam | ab192267 | 1:1000[a,b] |
| Rabbit monoclonal anti-NDUFB11 | Abcam | ab183716 | 1:1000[a,b] |
| Mouse monoclonal anti-NDUFA9 | Mol. Probes | 459100 | 1:1000[a] 1:2500[b] |
| Mouse monoclonal anti-NDUFS3 | MitoSciences | MS112 | 1:1000[a,b] |
| Rabbit polyclonal anti-NDUFS2 | Abcam | ab96160 | 1:1000[a,b] |
| Rabbit polyclonal anti-NDUFS1 | Proteintech | 12444-1-AP | 1:1000[a,b] |
| Rabbit polyclonal anti-NDUFV1 | Proteintech | 11238-1-AP | 1:1000[a,b] |
| Rabbit polyclonal anti-NDUFV2 | Proteintech | 15301-1-AP | 1:1000[a,b] |

[a]Dilution for SDS-PAGE/WB.
[b]Dilution for BN-PAGE/WB.

native polyacrylamide gel electrophoresis (BN-PAGE) western blot analysis as described below.

**SDS-PAGE western and blot analysis.** Total protein extracts from cultured cells were obtained with the RIPA buffer supplemented with protease inhibitor cocktail (Sigma) followed by sonication and clearance of the lysates. Protein concentrations were determined with Bradford reagent, and equal amounts of proteins were resuspended in a Laemmli buffer supplemented with beta-mercapthoethanol or TCEP. Isolated mitochondria were directly lysed in a Laemmli buffer. Samples were subjected to tris-glycine SDS-PAGE electrophoresis followed by the wet transfer on nitrocellulose membrane. Blots were decorated with antibodies followed by bio-luminescence detection with ECL reagent (Amersham). Densitometry-based quantification of western blots was performed with Image J Software. All original, uncropped blots can be found in the Source Data file. Information on antibody sources and dilutions can be found in Table 3.

**Blue native electrophoresis.** Mitochondrial protein concentrations were determined with Bradford reagent (Sigma). Equal amounts of mitochondria (15–25 µg) were lysed for 15 min on ice in dodecylmaltoside (DDM; 5 g/g of protein), and cleared from insoluble material for 20 min at $20,000 \times g$, 4 °C. Lysates were

combined with Coomassie G-250 (0.25% final). Mitochondrial complexes were resolved with BN-PAGE using the 4–16% NativePAGE Novex Bis-Tris Mini Gels (Invitrogen) in a Bis-Tris/Tricine buffering system with cathode buffer initially supplemented with 0.02% G-250 followed by the 0.002% G-250. Separated mitochondrial complexes were transferred onto a polyvinylidene fluoride membrane using the wet transfer methanol-free system. Membranes were immunodecorated with indicated antibodies followed by ECL-based signal detection. All original, uncropped blots could be found in the Source Data file.

**Pulse-chase immunoprecipitation.** Wild type (+/+) and CLPP-deficient (−/−) MEFs were grown till the 80% confluency in a high-glucose complete DMEM medium. Cells were pre-incubated in a methionine depletion media [DMEM without methionine supplemented with: glucose (4.5 g/l), sodium pyruvate (100 µg/ml), glutamine (4 mM), dialyzed FBS (10%), penicillin/streptomycin] for 15 min at 37 °C. Metabolic labeling was performed for 30 min at 37 °C in DMEM medium containing 0.15 mCi $^{35}$S-methionine, glucose (4.5 g/l), sodium pyruvate (100 µg/ml), glutamine (4 mM), dialyzed FBS (10%), penicillin/streptomycin. Labeling medium was removed, cells were washed twice with fresh complete DMEM medium containing the non-radiolabeled methionine, and incubated at 37 °C for indicated times. Cells were collected, washed with PBS and lysed in 1% SDS, 50 mM

Tris-HCl pH 7.4, 5 mM EDTA, 10 mM DTT, 15 U/ml DNase I, supplemented with protease inhibitor cocktail (Sigma). Cell lysates were boiled at 95 °C for 5 min followed by shearing with a 26G needle. Lysates were diluted 10x and proteins were quantified with Bradford reagent. One milligram of protein extract was incubated with 2 μl of polyclonal anti-NDUFV2 antibody overnight at 4 °C with rotational mixing. Twenty-five microliters of protein G-coupled Dynabeads™ were equilibrated in 1x lysis buffer, combined with antibody pre-incubated protein extracts, and immunoprecipitated for 1 h at 4 °C with rotational mixing. Beads were washed on a magnet with 1x lysis buffer, and proteins were eluted in a reducing 1x Laemmli at 37 °C with vigorous agitation. Immunoprecipitates were analyzed with SDS-PAGE followed by autoradiography detection on X-ray films. Inputs were used as a loading control.

**Oxygen consumption and extracellular acidification rates**. In all, $2 \times 10^5$ wild type and (+/+) deficient (−/−) MEF, and $2.5 \times 10^5$ of HeLa cells were pre-plated on a Seahorse plate in a full DMEM (high-glucose medium) or MEM medium, respectively. Oxygen consumption (OCR) and extracellular acidification rate (ECAR) were recorded using a Seahorse XF96 Analyzer (Seahorse Bioscience) in a final DMEM media containing 10 mM galactose, 2 mM L-glutamine, 1 mM sodium pyruvate, 1x non-essential amino acids (NEAA), pH 7.4. Each measurement was performed over 3 min after 3 min mix period. Basal measurements were collected four times, followed by three measurements after addition of oligomycin (final concentration 1 μM), followed by three measurements after addition of Carbonyl cyanide-4- (trifluoromethoxy) phenylhydrazone (FCCP) (final concentration 0.75 μM), followed by three measurements after addition of rotenone and antimycin A (final concentration 0.5 μM each).

**siRNA-mediated knockdown of Clpx and Lonp1**. Murine cell lines were transfected with siRNA oligonucleotide duplex (*Clpx siRNA 1, Clpx siRNA 2, Lonp1* siRNA, or *control siRNA* (Ref # SR-CL000-005; Eurogentec), using the Lipofectamine RNAi Max reagent (Invitrogen) following the reverse transfection protocol provided by the manufacturer. Transfections were performed 48 h before the analysis. The efficiency of downregulation was assessed by western blot analysis.

**High-resolution complexome profiling**. Mitochondria from hearts from wild type (+/+) and CLPP-deficient (−/−) 22-week-old female mice were isolated, solubilized with 6 g digitonin/g protein and native complexes were separated by BN-PAGE[9]. Lanes were cut into 60 even slices that were smashed and subjected to in-gel digestion with trypsin. Peptides were separated by liquid chromatography and analyzed by online tandem mass spectrometry (LC-MS/MS) in a Q-Exactive mass spectrometer equipped with an Easy nLC1000 nano-flow ultra high-pressure liquid chromatography system (Thermo Fisher Scientific) at the front end. The mass spectrometer operated in positive ion mode switching automatically between MS and data-dependent MS/MS of the top 20 most abundant precursor ions. Full scan MS mode (400 to 1400 *m/z*) was set at a resolution of 70,000 m/μm with an automatic gain control target of $1 \times 10^6$ ions and a maximum injection time of 20 ms. Selected ions for MS/MS were analyzed using the following parameters: resolution 17,500 m/μm, automatic gain control target $1 \times 10^5$; maximum injection time 50 ms; precursor isolation window 4.0 Th. Only precursor ions of charge $z = 2$ and $z = 3$ were selected for collision-induced dissociation. Normalized collision energy was set to 30% at a dynamic exclusion window of 60 s. A lock mass ion ($m/z = 445.12$) was used for internal calibration. Mass spectrometry data were analyzed with MaxQuant (version 1.5.0.25). Spectra were matched against the human NCBI Reference Sequence Database with known contaminants added[9]. The limit for the false-discovery rate determined by target-decoy database search was set to 0.01. Database searches were done with 20 ppm mass tolerance for precursor ions and fragmented ions, respectively. Two missed cleavages were allowed for tryptic digestion. N-terminal acetylation and oxidation of methionine were set as dynamic modifications and cysteine carbamidomethylation as fixed modification. Intensity basd absolute quantification (iBAQ) values derived from the MaxQuant analysis were corrected for loading variation between different samples using the total iBAQ values of all detected proteins listed in MitoCarta 2.0. For each protein, gel migration profiles were created and normalized to the maximum abundance across all samples analyzed. Then the migration patterns of the identified proteins were hierarchically clustered by an average linkage algorithm with uncentered Pearson correlation distance measures. Heat maps of the resulting complexome profiles consisting of a list of proteins grouped according to the similarity of their migration profiles in blue native gel electrophoresis were generated by representing the normalized abundance in each gel slice by a three color gradient (black/yellow/red)[9]. Microsoft Excel was used for visualization and analysis.

**CLPP and CLXP association with CI**. Plasmids containing genes encoding the murine wild type and proteolytically inactive CLPP (*Clpp^S149A*) fused to the double FLAG tag were constructed and transiently overexpressed for 48 h under the control of CMV promoter in CLPP-deficient MEFs, As a negative control, cells transfected with the empty plasmid (pcDNA3.1, Invitrogene) were used[18]. Mitochondria were isolated, lysed in DDM, and subjected to BN-PAGE. Bands corresponding to the fully assembled Complex I defined with the help of the in gel CI

activity were cut out from the BN-PAGE, prepared for a mass spectrometry (MS) analysis as described above (see High-Resolution Complexome Profiling).

**pSILAC-mediated enrichment for newly synthesized peptides**. Triplicates of equal numbers of *Clpp* +/+ and −/− MEF cells were cultured in complete, high-glucose DMEM medium to 70% confluency. Cells were washed with PBS and switched to depletion medium for 30 min [DMEM without methionine, arginine, and lysine, supplemented with: glucose (4.5 g/l), sodium pyruvate (100 μg/ml), glutamine (4 mM), dialyzed FBS (10%), penicillin/streptomycin]. Depletion medium was replaced by "intermediate" (CLPP+/+: 84 μg/ml [$^{13}C_6$] L-arginine and 146 μg/ml [4,4,5,5-D$_4$] L-lysine (Silantes)), or "heavy" (CLPP −/−: 84 μg/ml [$^{13}C_6$, $^{15}N_4$]] L-arginine and 146 μg/ml [$^{13}C_6$,$^{15}N_2$] L-lysine (Silantes)) labeling medium [DMEM supplemented with: 0.1 mM L-azidohomoalanine (AHA), glucose (4.5 g/l), sodium pyruvate (100 μg/ml), dialyzed FBS (10%), penicillin/streptomycin], and cells were metabolic labeled for 30 min. Oppositely SILAC-labeled cells were combined and washed twice with PBS. Cells were lysed in 8 M urea, 100 mM Tris-HCl, pH 8 buffer supplemented with protease inhibitor cocktail (Sigma) using the Bioruptor instrument, and L-AHA-labeled newly synthesized proteins were enriched using the Click-iT® protein Enrichment Kit (Thermo) according to manufacturer instructions. Resin-bound proteins were reduced with 10 mM DTT (in 100 mM Tris-HCl pH 8, 1% SDS, 250 mM NaCl, 5 mM EDTA) for 15 min at 70 °C, followed by alkylation with 40 mM iodoacetamide for 30 min in dark. Resin-bound proteins were stringently washed with SDS buffer (1% SDS, 100 mM Tris-HCl, pH 8, 250 mM NaCl, 5 mM EDTA), urea buffer (8 M urea, 100 mM Tris-HCl, pH 8), 20% isopropanol, and 20% ACN. Trypsin digestion of proteins bound to the resin was performed overnight at 37 °C in a buffer containing 2.5 μg/ml trypsin, 100 mM Tris–HCl pH 8, 2 mM CaCl$_2$, 10% ACN. Peptides were washed out from the resin, diluted to the final ACN concentration of 2%, acidified with formic acid (0.5% final), purified on SDB-RP StageTip discs and subjected to mass spectrometry.

All samples were analyzed on a Q-Exactive Plus Orbitrap (Thermo Scientific) mass spectrometer that was coupled to an EASY nLC (Thermo Scientific). Peptides were loaded with solvent A (0.1% formic acid in water) onto an in-house packed analytical column (50 cm—75 μm I.D., filled with 2.7 μm Poroshell EC120 C18, Agilent) and separated with 60 min or 150 min gradients. The mass spectrometer was operated in data-dependent acquisition mode and MS1 survey scan was acquired from 300 to 1750 *m/z* at a resolution of 70,000. The top ten most abundant peptides were isolated within a 1.8 Th window and subjected to HCD fragmentation using a collision energy of 27%. The AGC target was set to 5e5 charges, allowing a maximum injection time of 60 ms. Product ions were detected in the Orbitrap at a resolution of 17,500. Precursors were dynamically excluded for 20.0 s.

Mass spectrometric raw data were processed with Maxquant (version 1.5.3.8) using default parameters[47]. Briefly, MS2 spectra were searched against the Uniprot MOUSE fasta (downloaded at: 16.6.2017) database, including a list of common contaminants. False-discovery rates on protein and PSM level were estimated by the target-decoy approach to 1% (Protein FDR) and 1% (PSM FDR), respectively. The minimal peptide length was set to seven amino acids and carbamidomethylation at cysteine residues was considered as a fixed modification. Oxidation (M) and Acetyl (Protein N-term) were included as variable modifications. However, for the searches for oxidation products up to three variable modifications were used in multiple searches[48]. In Searches for cysteine oxidation products carbamidomethylation was treated as variable modification. The match between runs option was enabled. When SILAC labeling quantification was used, and the re-quantify option was enabled.

**Exchange rates of CI subunits**. Murine myoblast cell line C2C12 (CRL-1772) were cultured in medium containing DMEM, 10% fetal calf serum, 1 mM pyruvate, 4.5 g/l glucose, 2 mM glutamine, and 1% penicillin/streptomycin. For differentiation into contractile myotubes, confluent dishes were further cultured for 7 days in differentiation medium (DMEM with glutamax, 2% horse serum, 1 mM pyruvate, 4.5 g/l glucose, 2 mM glutamine and 1% penicillin/streptomycin). For pSILAC metabolic labeling, the medium was changed into differentiation medium containing $^{13}C_6$,$^{15}N_4$-L-Arginine and $^{13}C_6$,$^{15}N_2$-L-Lysine (Silantes). After 6 and 7 h cells were washed with PBS, collected by scraping and centrifugation. Cells were homogenized using a pre-cooled motor-driven glass/Teflon Potter-Elvehjem homogenizer at 1300 rpm and 40 strokes in 83 mM sucrose, 3.3 mM Tris/HCl, pH 7, 0.3 mM EDTA, 1.7 mM 6-aminohexanoic acid. Homogenates were centrifuged for 5 min at 500 x *g* to remove nuclei, cell debris, and intact cells. Mitochondria were pelleted by centrifugation for 10 min at 10,000× *g*.

Wild type (+/+) and CLPP-deficient (−/−) MEFs were cultured till 100% confluency in a complete DMEM medium (10% FBS, 4.5 g/l glucose, 1 mM pyruvate, 2 mM glutamine and penicillin/streptomycin). Cells were washed with a pre-warmed PBS and switched to the SILAC labeling medium [L-arginine/L-lysine free DMEM supplemented with $^{13}C_6$,$^{15}N_4$-L-Arginine, $^{13}C_6$,$^{15}N_2$-L-Lysine (Silantes), 10% dialyzed FBS, 4.5 g/l glucose, 1 mM pyruvate, 2 mM glutamine, and penicillin/streptomycin). After 5 h cells were washed with PBS and collected by scraping. Cells were homogenized using a pre-cooled homogenized with a rotating Teflon potter (40 strokes, 1200 rpm) in 83 mM sucrose, 3.3 mM Tris/HCl, pH 7, 0.3 mM EDTA,1.7 mM 6-aminohexanoic acid. Homogenates were centrifuged for 5 min at 850 x *g* to remove nuclei, cell debris, and intact cells. Mitochondria were pelleted by centrifugation for 5 min at 10,000× *g*.

Mitochondrial fraction from 20 mg C2C12 myotubes or 400 µg mitochondria from MEFs were resuspended in 35 µl solubilization buffer (50 mM imidazole pH 7, 50 mM NaCl, 1 mM EDTA, 2 mM aminocaproic acid) and solubilized with 10 µl 20% digitonin (Se rva) and centrifuged for 20 min at $22,000 \times g$. Supernatants were supplemented with 2.5 µl 5% Coomassie G-250 in 500 mM aminocaproic acid and 5 µl 0.1% Ponceau S in 50% glycerol. Equal protein amounts of samples were loaded on top of a 3 to 18% acrylamide gradient gel (dimension $14 \times 14$ cm). After native electrophoresis in a cold chamber, blue native gels were fixed in 50% (v/v) methanol, 10% (v/v) acetic acid, 10 mM ammonium acetate for 30 min and stained with Coomassie (0.025% Serva Blue G, 10% (v/v) acetic acid).

Each lane of a BN-PAGE gel was cut into 48 equal fractions and collected in 96-filter well plates (30–40 µm PP/PE, Pall Corporation). The gel pieces were de-stained in 60% Methanol, 50 mM ammoniumbicarbonate (ABC). Solutions were removed by centrifugation for 2 min at $600 \times g$. Proteins were reduced in 10 mM DTT, 50 mM ABC for 1 h at 56 °C and alkylated for 45 min in 30 mM iodoacetamide. Samples were digested for 16 h with trypsin (sequencing grade, Promega) at 37 °C in 50 mM ABC, 0.01% Protease Max (Promega) and 1 mM CaCl₂. Peptides were eluted in 30% ACN and 3% formic acid, centrifuged into a fresh 96-well plate, dried in speed vac and resolved in 1% acetonitrile and 0.5% formic acid.

### BIAM-assay and label-free quantification of cardiac mitochondrial proteome.
Mitochondria from hearts of three wild type (+/+) and three CLPP-deficient (−/−) animals (20-week-old; males) were isolated and incubated for 10 min with 50-fold molar excess of a N-ethylmaleimide (NEM)/cysteines ratio in buffer consisting of 100 mM sucrose, 100 mM KCl, 1 mM EDTA, 20 mM TES, pH 7.2 at room temperature to block free cysteine thiols. Mitochondria were again washed with NEM containing buffer. Mitochondrial pellets containing 300 µg protein were resuspended in 100 µl DTT-DB (8 M Urea, 5 mM EDTA, 0.5% SDS, 50 mM Tris/HCL, pH 8.5, 2 mM DTT). One-hundred microliters of biotinylated iodoacetamide (BIAM)-DB (8 M Urea, 5 mM EDTA, 0.5% SDS, 50 mM Tris/HCL, pH 8.5, 50x molar excess BIAM (EZ-Link™ Iodoacetyl-PEG2-Biotin, ThermoFisher Scientific) /Cysteine) was added and samples were incubated for 60 min at 22 °C in the dark. Protein solutions were divided into two equal parts (one for BIAM-Assay and one for label-free quantitative proteomics) and were precipitated with ice-cold acetone overnight at −20 °C, collected by centrifugation. BIAM samples were resuspended in 100 µl lysis buffer (5 mM EDTA, 50 mM Tris/HCl pH 8.5, 1% Triton-X-100, 1% SDS). Oxidized proteins that now contain a BIAM label on cysteines were affinity purified using agarose streptavidin beads (Thermo Scientific) overnight at 4 °C on a wheel. After washing, beads (BIAM-assay) and protein pellet (label-free proteome) were resuspended in 50 µl 6 M guanidine hydrochloride (GdmCl), 50 mM Tris/HCl, pH 8.5, respectively, and incubated at 95 °C for 5 min. Samples were diluted with 25 mM Tris/HCl, pH 8.5, 10% ACN to obtain a final GdmCl concentration of 0.6 M. Proteins were digested with 1 µg Trypsin (sequencing grade, Promega) overnight at 37 °C under gentle agitation. Digestion was stopped by adding trifluoroacetic acid to a final concentration of 0.5%. Peptides were loaded on multi-stop-and-go tip (StageTip)[49]. Three fractions from SCX stage tips were collected in wells of microtiter plates and peptides were dried and resolved in 1% ACN, 0.1% formic acid.

### Mass spectrometry for pSILAC complexome and BIAM-assay.
Liquid chromatography/mass spectrometry (LC/MS) was performed on Thermo Scientific™ Q-Exactive Plus equipped with an ultra-high performance liquid chromatography unit (Thermo Scientific Dionex Ultimate 3000) and a Nanospray Flex Ion-Source (Thermo Scientific). Peptides were loaded on a C18 reversed-phase precolumn (Thermo Scientific) followed by separation on a with 2.4 µm Reprosil C18 resin (Dr. Maisch GmbH) in-house packed picotip emitter tip (diameter 100 µm, 15 cm from New Objectives) using a gradient from 4% ACN, 0.1% formic acid to 40% eluent B (99% acetonitrile, 0.1% formic acid) for 30 min followed by a second gradient to 60% B with a flow rate 400 nl/min and washout with 99% B for 5 min. The gradient for fractions to identify oxidized proteins was to 30% B for 90 min followed by a second gradient to 60% B for additional 15 min.

MS data were recorded by data-dependent acquisition. The full MS scan range was 300 to 2000 $m/z$ with a resolution of 70,000, and an automatic gain control (AGC) value of $3*10^6$ total ion counts with a maximal ion injection time of 160 ms. Only higher charged ions (2+) were selected for MS/MS scans with a resolution of 17,500, an isolation window of 2 $m/z$ and an automatic gain control value set to $10^5$ ions with a maximal ion injection time of 150 ms. MS1 Data were acquired in profile mode.

For data analysis MaxQuant v1.6.0.1 (BIAM-Assay and C2C12 pSILAC complexome) and v1.6.1.0 (pSILAC Complexome of MEFS)[50], Perseus 1.5.6.0[51] and Excel (Microsoft Office 2013) were used. Proteins were identified with mouse reference proteome database UniProtKB with 52538 entries, released in 2/2018. Acetylation (+42.01) at N-terminus and oxidation of methionine (+15.99) were selected as variable modifications and carbamidomethylation (+57.02) as a fixed modification on cysteines. For the BIAM-Assay, additional variable modification on cysteines but no fixed modification were selected: NEM (+125.05) and BIAM (+414.19). The enzyme specificity was set to Trypsin. False-discovery rate (FDR) for the identification protein and peptides was 1%.

For pSILAC complexomes, intensity-based absolute quantification (IBAQ) values were recorded. Heatmap of proteins containing light isotopes represents the abundance normalized to a maximum appearance in a BN-PAGE lane. Protein abundance of newly translated heavy isotopes was normalized to the maximum

appearance of light labeled proteins. Slice the number of the maximum appearance of mouse mitochondrial complex III dimer (483,272 Da), Complex IV (213,172 Da), complex V (537,939 Da), complex I (979,577 Da) and respiratory supercomplex containing complex I, III dimer and one copy of complex IV (1,676,021 Da) was used for native mass calibration.

For the BIAM-Assay and quantitative proteome analysis of heart mitochondria, reverse identifications and common contaminants were removed, and the data-set was reduced to proteins that were identified in 2 of 3 samples in at least one experimental group. Missing values were replaced by the background from the normal distribution. Oxidized proteins were normalized to the mean of protein abundance. Differentially oxidized proteins were determined by permutation-based false-discovery rate (FDR) calculation and student´s t-test.

### Electron paramagnetic resonance analysis.
To obtain mitochondrial membranes for EPR analysis, the hearts from 20- to 26-week-old animals were homogenized with a rotating Teflon potter (Potter S, Sartorious; 20 strokes, 1000 rpm) in a buffer containing 100 mM sucrose, 50 mM KCl, 20 mM TES, 0.2% fatty acid-free BSA, subtilisin (1 mg per 1 g of tissue), pH 7.6). Mitochondria were isolated by differential centrifugation (850 x g, followed by 8500 x g; at 4 °C), washed twice in BSA-free buffer, and permeabilized on ice with gentle agitation for 15 min in 0.1% digitonin, 100 mM sucrose, 50 mM KCl, 20 mM TES, pH 7.2. Membranes were pelleted at $10,000 \times g$ 4 °C, and washed twice in 100 mM sucrose, 50 mM KCl, 20 mM TES, pH 7.6, and snap-freeze in liquid N₂. Mitochondrial membranes (10–15 mg protein per ml) were incubated with NADH (2 mM, final concentration) at ambient temperature, transferred into an EPR tube and frozen in isopentane: methyl cyclohexane (5:1, v–v) cooling mixture (approx. 120 K). Samples were stored in liquid nitrogen until EPR analysis. Low-temperature cw-EPR spectra were obtained using a Bruker ESP 300E spectrometer equipped with a liquid helium continuous flow cryostat, ESR 900 (Oxford Instruments) using the following instrument settings: microwave frequency 9,47 GHz, modulation amplitude 1 mT, conversion time 82 ms, time constant 82 ms, microwave power 1 mW, temperature 12 K. After EPR spectroscopy, samples were thawed and CI was analyzed by native gel electrophoresis, in gel activity staining, and western blotting.

### Mitochondrial protein synthesis in cells.
Wild type (+/+) and CLPP-deficient (−/−) MEF, HEK293T, and HeLa cells were grown till 90% confluency in respective media. Cells were switched to depletion medium [DMEM without methionine supplemented with: glucose (4.5 g/l), sodium pyruvate (100 µg/ml), glutamine (4 mM), dialyzed FBS (10%), penicillin/streptomycin, emetine (100 µg/ml) for 15 min at 37 °C. Metabolic labeling of mitochondrially encoded subunits was performed in the presence of irreversible inhibitor of cytoplasmic translation (emetine 100 µg/ml) for 60 min at 37 °C in DMEM medium containing 0.15 mCi ³⁵S-methionine, glucose (4.5 g/l), sodium pyruvate (100 µg/ml), glutamine (4 mM), dialyzed FBS (10%), penicillin/streptomycin. Cells were washed with complete DMEM medium containing methionine, and incubated at 37 °C for 5 more minutes. Cells were collected, washed with PBS, and lysed in a RIPA buffer supplemented with protease inhibitor cocktail (Sigma) followed by shearing through the 26G needle. Protein concentration was determined with Bradford reagent, and 50 µg of proteins was subjected into SDS-PAGE. Gels were stained with Instant Blue™ Ultrafast Protein Stain (Sigma), dried and exposed to X-ray films.

### Drug-mediated inhibition of mitochondrial protein synthesis.
The CI decay and recovery were monitored in wild type (+/+) and CLPP-deficient (−/−) MEFs. Cells at the 70% confluency were exposed to the inhibitors of mitochondrial translation (doxycycline 15 µg/ml; chloramphenicol 200 µg/ml) and incubated up to 48 h. After that, cells were transferred to the rug-free medium for the next 48 h (chloramphenicol). At indicated time points, mitochondria were isolated from cells and subjected to BN-PAGE western blot analysis.

### Mitochondrial import assay.
Murine NDUFS3 cDNA clone (IRAVp968H0352D) and NDUFV2 cDNA clone (Riken: 6530404K08) were purchased from Source Bioscience and subcloned into Luciferase SP6 control DNA (Promega). Radiolabeled proteins were synthesized in the presence of ³⁵S-methionine using the TnT® Quick Coupled Transcription-Translation Rabbit Reticulocyte System (Promega) accordingly to manufacturer instructions. Mitochondria from Clpp+/+ and Clpp−/− MEFs were isolated and quantified with Bradford reagent. Fifty micrograms of mitochondria were resuspended in 50 µl of import buffer (20 mM HEPES–KOH pH 7.4, 250 mM sucrose, 80 mM KOAc, 5 mM MgOAc, 10 mM sodium succinate, 5 mM ATP, and 5 mM methionine). In control assays, the membrane potential was dissipated with 10 µM valinomycin. After the addition of 5 µl of lysate containing the radiolabelled precursor, samples were incubated at 37 °C for indicated times. Mitochondria were pelleted (10,000 x g, 5 min, 4 °C), and resuspended in 50 µl of buffer containing 20 mM HEPES–KOH, pH 7.0, 500 mM sucrose). Where indicated, samples were treated with proteinase K (50 µg/ml) for 15 min, and the reaction was quenched with 1 mM phenylmethylsulfonyl fluoride. Proteins were precipitated with trichloroacetic acid (TCA), washed with acetone, solubilized in 1x Laemmli buffer, and subjected to SDS-PAGE. Protein import was determined on autoradiograms obtained from dried gels exposed to X-ray films.

For the incorporation of imported subunits into CI, cardiac mitochondria were isolated from 20- to 24-week-old animals and quantified with Bradford reagent. Twenty-five micrograms of mitochondria were resuspended in 50 µl of import buffer (250 mM sucrose, 5 mM MgAc, 80 mM KAc, 20 mM HEPES–KOH pH 7.4, 10 mM sodium succinate, 5 mM ATP, 1 mM DTT). Five microliters of reticulocyte lysate containing the radiolabelled CI subunit precursor was added into import reactions and incubated at 37 °C with mild agitation for the indicated time. Mitochondria were pelleted (4 °C, 12,000 x $g$, for 5 min), resuspended in DDM (5 g/g protein), and subjected to BN-PAGE. Autoradiograms were obtained from the dried gels exposed to X-ray films.

**De novo synthesis and assembly of mtDNA-encoded subunits.** Wild type (+/+) and CLPP-deficient (−/−) MEFs were grown till 70% confluency in a complete, high-glucose DMEM medium. Cells were switched to depletion medium [DMEM without methionine supplemented with: glucose (4.5 g/l), sodium pyruvate (100 µg/ml), glutamine (4 mM), dialyzed FBS (10%), penicillin/streptomycin, cycloheximide 200 µg/ml] for 15 min at 37 °C. Metabolic labeling of mitochondrially encoded proteins was performed in the presence of cycloheximide (200 µg/ml) for 30 min at 37 °C in DMEM medium containing 0.15 mCi $^{35}$S-methionine, glucose (4.5 g/l), sodium pyruvate (100 µg/ml), glutamine (4 mM), dialyzed FBS (10%), penicillin/streptomycin. Cells were washed twice with complete DMEM medium containing non-radiolabelled methionine and incubated at 37 °C for the indicated time. Cells were collected, washed with PBS, and resuspended in a digitonin buffer (2 mg/ml in PBS). After 10 min of incubation on ice, the cell suspension was 6x diluted with PBS, and crude mitochondria were pelleted for 5 min, 10,000 x $g$, 4 °C. Mitochondria were washed twice with cold PBS, resuspended in DDM (5 g/g protein), and proteins separated by BN-PAGE. Autoradiograms were obtained from the dried gels exposed to X-ray films.

**Inverse redox shift assay.** Cells were grown on 6-well plates till 70% confluency. Cells were pre-treated with drugs or transfected with siRNA as indicated in respective figures. For minimum shift and maximum shift sample, cells were washed with PBS. For steady-state samples and drug pre-treated samples, cells were washed and incubated with ice-cold 20 mM NEM for 10 min on ice. NEM-PBS was removed, and cells were collected in ice-cold 8% TCA. Proteins were precipitated in TCA overnight at −20 °C, and pelleted for 15 min at 20,000 x $g$, 4 °C. Proteins pellets were washed with ice-cold 5% TCA and solubilized in 1x Laemmli buffer containing 10 mM tris(2-carboxyethyl)phosphine (TCEP) with help of sonication. Samples were incubated for 15 min at 45 °C with vigorous agitation and cooled down at room temperature. Minimum shift samples were modified with 15 mM NEM. Maximum shift, steady-state and drug pre-treated samples were modified with 15 mM mm(PEG)$_{24}$. Samples were incubated at room temperature in the dark for 1 h. Redox state was analyzed by SDS-PAGE western blotting.

**N-module accessibility upon differential protein extraction.** Cardiac heart mitochondria from wild type (+/+) and CLPP-deficient (−/−) 20-week-old animals were isolated and protein concentration was determined with Bradford reagent. One-hundred micrograms of mitochondria were resuspended in (a) 2 M NaCl in PBS and extracted by vigorous vortex for 5 min; (b) 1% digitonin (w/v) in PBS and lysed on ice for 15 min with occasional vortexing; (c) 1% Triton-X-100 in PBS and lysed on ice for 15 min with occasional vortexing. Lysates were spun and 20,000 x $g$, 15 min, 4 °C. Supernatants were collected and incubated with 2 µl of anti-NDUFV2 antibody overnight at 4 °C with rotational mixing. Thirty microliters of protein G-coupled Dynabeads$^{TM}$ were equilibrated in respective lysis buffer, combined with antibody pre-incubated protein extracts, and immunoprecipitated for 1 h at 4 °C with rotational mixing. Beads were washed on a magnet with respective lysis buffer, and proteins were eluted in a reducing 1x Laemmli at 37 °C with vigorous agitation. Immunoprecipitates were analyzed with SDS-PAGE western blotting and input mitochondria were used as a loading control.

**In vivo determination of the ROS levels.** HeLa wild type (+/+) and CLPP-deficient (−/−) cells were cultured in DMEM (Sigma) containing 10% fetal calf serum (FCS, Biowest) and 500 µg/ml Penicillin/Streptomycin (Sigma) at 37 °C and 5% CO$_2$. For imaging experiments, cells were seeded in Poly-L-Lysine (Sigma)-coated 96-well plates (Greiner Bio-One). Twenty-four hours after seeding, cells were transfected with 0.05 µg DNA (mt-HyPer3/pcDNA5 or mt-SypHer/pC1, respectively) mixed with 0.4 µl FuGENE HD (Promega) in 10 µl DMEM per well.

Fluorescence of mt-HyPer3 and mt-SypHer was assessed 48 h after transfection, using the Cytation3 cell multi-mode reader (BioTek) with a 10x in-air objective. Transfected cells were washed once and then incubated in 50 µl minimal medium (140 mM NaCl, 5 mM KCl, 1 mM MgCl$_2$, 2 mM CaCl$_2$, 20 mM HEPES, 10 mM glucose, adjusted to pH 7.4 with NaOH) for 30 min at 37 °C and 5% CO$_2$ inside the reader. For excitation of mt-HyPer3 and mt-SypHer, the BioTek LED filter cubes 390 ± 9 nm and 469 ± 17 nm were used. Emission was in both cases detected at 525 ± 19 nm. Cells were first left unperturbed and analyzed for 5 min in 1 min intervals ("steady state"). Then, the response to H$_2$O$_2$ was analyzed by addition of 30 µl H$_2$O$_2$ (different stock solutions; automatic injection) to final concentrations between 0 and 55 µM H$_2$O$_2$. Recording intervals were 30 s. The images were analyzed using RRA

("redox ratio analysis" software[52] and R. Statistical analysis: Shapiro–Wilk test to test for normal distribution followed by Mann–Whitney-$U$-test.

**Analysis of FMN content.** Mitochondria isolated from cultures cells from several independent experiments ($n$ = 3–6) were pooled and further pursued as technical replicates. For membrane permeabilization, mitochondria were incubated on ice in a presence of 80 µg/ml alamethicin for 15 min with occasional vortexing. Permeabilization was followed by two washings with mitochondria isolation buffer. FMN content was determined using the modified protocol[24,53]. In brief, 100 µg of intact mitochondria or mitochondrial membranes were resuspended in 100 µl of water and combined with equal volume of 15% TCA. Samples were incubated on ice for 10 min, and deproteinized by centrifugation at 10,000 x $g$ for 10 min, 4 °C. A 1:10 volume of 4 M K$_2$HPO$_4$ (pH unadjusted) was added to partially neutralize the supernatant (to pH ~2.5). Two-hundred microliters of sample was loaded on a 96-well plate. The fluorescence was recorded using the EnSpire plate reader (Perkin Elmer) at excitation/emission 450/525 nm, auto PMT, ten readings. First, the fluorescence was recorded at acidic conditions, and then under neutral pH conditions (pH ~ 7.4) obtained by addition of 30 µl of 4M K$_2$HPO$_4$. Freshly prepared FMN standard solutions were used to calibrate the fluorescence signal.

**Reporting summary.** Further information on research design is available in the Nature Research Reporting Summary linked to this article.

## Data availability

The authors declare that all data supporting the findings of this study are available within the article and its Supplementary Information files or from the corresponding author upon reasonable request. The mass spectrometry proteomic data have been deposited in the Proteome Exchange Consortium within the "PRIDE" partner repository under the following accession codes: PXD014897; PXD017463; PXD017464; PXD017465; PXD017614.

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

## Acknowledgements

We wish to thank Jose Antonio Enriquez for L929 Cytb mutant and control cells, and Cox10 KO and control cells. We thank Jana Meisterknecht for excellent technical assistance for native electrophoresis and sample preparation for mass spectrometry. We thank Elena Rugarli (University of Cologne) and Thomas Langer (MPI-AGE) for reading the paper and providing us with a critical input. The study was supported by Aleksandra Trifunovic's grants of the European Research Council (ERC- StG-2012-310700 and ERC-2018-PoC- 813169), Deutsche Forschungsgemeinschaft (DFG, German Research Fundation) - SFB 1218 - Projektnummer 26992540, and Center for Molecular Medicine Cologne, University of Cologne. This study was also supported by Ilka Wittig's funding provided by the Deutsche Forschungsgemeinschaft: SFB 815/Z1 and by the BMBF mitoNET—German Network for Mitochondrial Disorders 01GM1906D.

## Author contributions

Conceptualization: K. Szczepanowska, A.T., J.R., M. Herholz, U.B., I.W. Data curation: K. Szczepanowska, I.W., U.B., S.G., S.M., K. Zwicke, C.F. Formal analysis: K. Szczepanowska, A.T., I.W., U.B., H.G., S.M., C.F. Funding acquisition: A.T., I.W. Investigation: K. Szczepanowska, K. Senft, J.H., M. Herholz, A.K., M.N.H, E.H., C.B., S.K., K. Zwicke, S.G.C., L.B., J.K., A.R., C.F. Visualization: K. Szczepanowska, M. Herholz, I.W., U.B., M.N.H., A.T. Writing: K. Szczepanowska, A.T., U.B., J.R., I.W.

## Competing interests

The authors declare no competing interests.
