## [Peer Review File · Nature Communications]

Reviewers' comments:

Reviewer #1 (Remarks to the Author):

This manuscript reports the discovery of the novel regulatory pathway in mitochondria. The increased turnover of the N-module of complex I is shown to be mediated by mitochondrial matrix protease ClpXP. This seems to be a protection against accumulation of dysfunctional CI - N-module as an entry point for electrons into complex I is most damaged by ROS and perhaps just by wear and tear. Very large amount of well-presented data is reported, verifying the proposed mechanism. The manuscript is clearly written and well referenced, presenting a major discovery in mitochondrial biology, and so is suitable for publication in Nature Communications.

Reviewer #2 (Remarks to the Author):

This manuscript by Szczepanowska and colleagues reports on the role of mitochondrial proteases, specifically ClpXP on the repair and remodeling of mitochondrial respiratory chain complex I. Whereas a similar kind of repair has been reported for plastid photosynthetic enzymes exposed to UV radiation, this is the first time that such a mechanism is reported for mitochondrial respiratory enzymes. Regarding the complex I repair reported here, the complex module affected is the NADH-oxidising module, which could be more susceptible to oxidative damage. The authors, however, uncover an exceptional mechanism, involving thiol chemistry to modify specific subunits and inactivate the N-module that would protect the cell from ROS production upon OXPHOS malfunctioning. Another piece of important information relates to the overall stabilization of CI by depletion of ClpXP, in cases in which the assembly of complex III or IV is affected. The data presented add a new layer of regulation of CI stability, beyond its sole interaction with these complexes in supercomplexes. The manuscript is excellent technically and conceptually. Perhaps, the only point whose discussion should be tempered is the possible role of ClpXP as a possible therapeutic target for the treatment of mitochondrial diseases. Even if CI would be stabilized, it is not clear that this would lead to a more permanent activation of the enzyme; and furthermore, given the multiple substrates of the protease, unwanted side effects may surely arise.

Reviewer #3 (Remarks to the Author):

Overview

This is a very interesting paper that shows convincingly that the N-module of complex I - the site where electrons enter the complex from NADH - is turned over more rapidly than the rest of the large complex. The model that arises is of damage to complex I being repaired by recycling and replacement of particular damage sensitive modules with a key role for ClpXP in this. This model has great appeal and this work will go a long way to initiate the testing of its validity. However, while I am supportive of the paper as a whole there are a number of points that could be clarified. In parts I found the paper quite hard going and in some places I was not sure if I'd fully grasped the mechanism the authors were proposing - of course this may reflect my limitations as much as the clarity of the writing! I look forward to seeing what the other reviewers made of the manuscript.

Major points

1 The role to which ClpXP can act as a chaperone as well as a protease and also to encourage the insertion of new complex I subunits into the N-module was not clear to me. For example is the damaged N-module removed in its entirety or piecemeal? Would ClpXP act to insert the new N-module as a whole into the complex, or polypeptide by polypeptide? I'm aware that the answers to these may be unresolved but some clarity on the points of uncertainty would help the reader.

2 Can the authors comment on whether this system recognises and replaces damaged N modules, or if it minimises damage by the rapid recycling of this damage-prone module? Also is it possible to separate the removal of the N module with degradation of the polypeptides?

3 In Fig 1a seems to show the fastest exchange is in the Q module subunits, rather than in the N module?

4 In Fig 1 b is there any information on the background of peptides interacting with the inactive CLPXP that are from polypeptides from outside complex I? The data here seem a bit hard to interpret without some idea of the level of background binding.

5 In Fig 1c, are there control complex I polypeptides that were not affected by CLPP1 knock out that were not in the N module?

6 I don't follow how shear stress would be damaging to complex I and whether this is associated with loss of FMN content? Is it possible to assess FMN and FeS content of the free N modules? If the N module did contain an FMN I might expect this to produce ROS?

7 It should be possible to roughly assign the cys residues that are seen in Fig 4 to FeS centres, for example by considering the number of FeS numbers in this subunit and by calculating the stoichiometry of the Cys residues modified with the large molecular weight label. I may have missed something, but I wasn't clear how the authors could distinguish cys labelling of de novo synthesised subunits that were labelled before insertion of FeS centers, from damaged subunits that were removed from the complex with the cys residues becoming exposed as the FeS centers were stripped out?

8 The role of stalling of oxidative phosphorylation was unclear to me. Does stalling just mean slow respiration? Does this mean lowered respiration and a build-up of electrons in the respiratory chain? Does this imply that complex I is affected when electrons backed up produce ROS by reduction of the FMN on complex I, leading to damage to complex I? Is there any link between this process and the active/deactive transition in complex I?

9 Does the effect of CLPXP loss on zebra fish with unstable complex I correlate with an increase in complex I activity in isolated mitochondria? Or to changes in ATP/ADP ratio in the adult fish? Without some bioenergetic assessment I find it difficult to interpret these findings.

Reviewer #4 (Remarks to the Author):

Szczepanowska have evaluated complex I assembly. They have demonstrated that the mitochondrial protease ClpP degrades complex I subunits in order to promote optimal complex I assembly into superstructures.

The work involves intricate biochemical analysis and provides additional insights into the assembly of complex I and the function of ClpP.

The paper is very dense and the methods are not fully explained. For example, the authors conduct complexome profiling. The authors cite a previous paper regarding this assay, but provide no description of the assay in the results or methods. This example and many others make the

paper difficult to understand and follow. In many places, the sentences are not well linked, so data is hard to understand. Often figures are cited without an explanation of the results. As such, I found the paper to be highly specialized. Perhaps with better writing it might be more accessible to a broader audience.

The authors use ClpP knockouts and knockdowns, but don't show levels of ClpP in these cells. These data would be important controls to demonstrate changes in target levels.

The authors use siRNA to knockdown targets like ClpX. In all of these cases, the immunoblot should be shown to demonstrate target knockdown. In addition, more than 1 siRNA or a rescue experiment should be performed to support on-target effects.

Response to Editor comments:

We ask that you respond to all of the queries from Reviewer #3, including data on background peptide binding to ClpXP and a bioenergetic assessment of ClpXP loss in zebrafish. We also ask that you perform the siRNA validation and controls suggested by Reviewer #4 and rewrite the paper to improve clarity (Reviewers #3 and #4)

In the revised manuscript we have addressed all concerns raised by reviewers, and rewrote the manuscript to improve clarity.

Point-by-point detailed answers to questions raised by the reviewers

We thank the reviewers for their helpful suggestions to improve our paper and their overall positive comments. Our response to the reviewers is as follows:

Reviewer 1:

This manuscript reports the discovery of the novel regulatory pathway in mitochondria. The increased turnover of the N-module of complex I is shown to be mediated by mitochondrial matrix protease ClpXP. This seems to be a protection against accumulation of dysfunctional CI - N-module as an entry point for electrons into complex I is most damaged by ROS and perhaps just by wear and tear. Very large amount of well-presented data is reported, verifying the proposed mechanism. The manuscript is clearly written and well referenced, presenting a major discovery in mitochondrial biology, and so is suitable for publication in Nature Communications.

We thank the reviewer for the very positive evaluation of our manuscript.

Reviewer 2:

This manuscript by Szczepanowska and colleagues reports on the role of mitochondrial proteases, specifically ClpXP on the repair and remodeling of mitochondrial respiratory chain complex I. Whereas a similar kind or repair has been reported for plastid photosynthetic enzymes exposed to UV radiation, this is the first time that such a mechanism is reported for mitochondrial respiratory enzymes. Regarding the complex I repair reported here, the complex module affected is the NADH-oxidising module, which could be more susceptible to oxidative damage. The authors, however, uncover an exceptional mechanism, involving thiol chemistry to modify specific subunits and inactivate the N-module that would protect the cell from ROS production upon OXPHOS malfunctioning. Another piece of important information relates to the overall stabilization of CI by depletion of ClpXP, in cases in which the assembly of complex III or IV is affected. The data presented add a new layer of regulation of CI stability, beyond its sole interaction with these complexes in supercomplexes. The

manuscript is excellent technically and conceptually. Perhaps, the only point whose discussion should be tempered is the possible role of ClpXP as a possible therapeutic target for the treatment of mitochondrial diseases. Even if CI would be stabilized, it is not clear that this would lead to a more permanent activation of the enzyme; and furthermore, given the multiple substrates of the protease, unwanted side effects may surely arise.

We thank the reviewer for the overall positive evaluation of our manuscript. We do agree with the point raised by the reviewer. It is possible that in cases where primary defect specifically impairs CI function, stabilization of CI does not actually provide beneficial effect by increasing CI function. Instead, increased levels of CI might lead to better preservation of cristae, and therefore mitochondrial morphology due to increased stability of supercomplexes. Alternatively, presence of functional free N-module that maintains NAD⁺/NADH ratio might also be very beneficial for mitochondria. We have clarified this better in the Discussion of the revised manuscript. In addition, ClpXP is a multifaceted protease, with potential tissue-specificity; therefore, its use as a target in therapies requires particular caution and rigorous testing. Nevertheless, we have shown that loss of ClpP can be beneficial in heart specific deficiency of mitochondrial aspartyl-tRNA synthase. Although we initially proposed that the beneficial effect stems from the improvement of mitochondrial translation, with these new results, we think that stabilization of CI might also be a beneficial factor in this model. What is even more remarkable, we showed that this effect is dose-dependent, making the therapeutic application even more plausible (Seiferling et al, 2016).

Reviewer 3:

This is a very interesting paper that shows convincingly that the N-module of complex I - the site where electrons enter the complex from NADH – is turned over more rapidly than the rest of the large complex. The model that arises is of damage to complex I is being repaired by recycling and replacement of particular damage sensitive modules with a key role for ClpXP in this. This model has great appeal and this work will go a long way to initiate the testing of its validity. However, while I am supportive of the paper as a whole there are a number of points that could be clarified. In parts I found the paper quite hard going and in some places I was not sure if I'd fully grasped the mechanism the authors were proposing – of course this may reflect my limitations as much as the clarity of the writing! I look forward to seeing what the other reviewers made of the manuscript.

We thank the reviewer for the overall positive assessment of our manuscript. We now rewrote parts of the manuscript to make it more understandable and have highlighted changes in the revised manuscript. In addition, we have added schemes in some critical figures that should help clarify our results.

1. The role to which ClpXP can act as a chaperone as well as a protease and also to encourage the insertion of new complex I subunits into the N-module was not clear to me. For example is the damaged N-module removed in its entirety or piecemeal? Would ClpXP act to insert the new N-module as a whole into the complex, or polypeptide by polypeptide? I'm aware that the answers to these may be unresolved but some clarity on the points of uncertainty would help the reader.

Indeed, our results argue that ClpXP might play a dual role in N-module dynamics, facilitating both its insertion and removal from CI. Although in the absence of CLPP the inactivated N-module remains attached to the CI, depletion of the whole ClpXP complex causes detachment of N-module from CI, arguing that the CLPX chaperone function might stabilize the inactive CI (Figure 4a). We don't know whether the N-module is removed in pieces or entirely. All three subunits seem to simultaneously disappear from CI, in conditions that induce loss of N-module, allowing us to speculate that core N-module is possibly taken away as a whole subcomplex. However, more dynamic measurement would be needed to fully confirm this. Our results also show that, in CLPP deficient hearts, free N-module accumulates in distinct subcomplexes that contain all four subunits that form functional N-module (V1, V2, S1 and A6), arguing that N-module is pre-formed in the mitochondrial matrix and added in the form of subcomplex directly to the rest of CI. This is also in agreement with previously published model for the CI assembly (Guerrero-Castillo S et al. 2017). However, we also showed that in highly proliferating cells, the exchange of the V1 and V2 on preexisting CI is slower than the exchange of S1 subunit, when CLPP is not present. This suggests that depending on the tissue-specific metabolic demand, and even stressor, N-module turnover might be different. Our results also argue against presence of the rest of accessory subunits in the free N-module intermediates in both heart and proliferating tissues.

2 Can the authors comment on whether this system recognizes and replaces damaged N modules, or if it minimizes damage by the rapid recycling of this damage-prone module? Also is it possible to separate the removal of the N module with degradation of the polypeptides?

We think that CLPXP recognizes some specific structural features appearing at the surface of the damaged/inactivated N-module, and this event will propagate the exchange of the affected subunits and the entire module. However, with our current knowledge, we cannot claim whether CLPP recognizes any kind of N-module damage or just complete N-module inactivation. It is possible that a minor N-module damage that precedes the N-module inactivation will already be enough to recruit the ClpXP. We think that is not possible to separate the removal of N-module from its degradation in the normal conditions. However, we cannot

exclude that specific stressors/environmental triggers in mitochondrial matrix milieu can modulate the intrinsic ClpXP activity. This would then indicate that ClpXP could be also able to act before the actual damage to N-module happens. Indeed, some of our findings indicate that CLPX chaperone activity may partially stabilize the N-module on CI when CLPP is absent (Figure 4a). This would suggest that two events, the substrates recognition and degradation, can be spatially and temporary separated, at least upon certain conditions.

3 In Fig 1a seems to show the fastest exchange is in the Q module subunits, rather than in the N module?

In differentiated C2C12 myoblasts the fastest exchange is observed for the several accessory N- and Q-module subunits (S4, A6, A7, S6), which form a kind of collar around the adjacent parts of N- and Q-module to bind them together. Those subunits are the last ones to assemble to the fully formed CI together with pre-formed core N-module (V1, V2, S1 and A2) according to previously published model (Guerrero-Castillo S et al. 2017). However, the exchange rate of these subunits is not changed upon the CLPP deficiency, and some even show mildly increased turnover in CLPP deficient cells. This suggests that N-module damage/inactivation might cause conformational change that might be sensed on the level of these subunits, which are then exchanged by action of other mitochondrial proteases, allowing the access of CLPP to the core N-module part.

4 In Fig 1 b is there any information on the background of peptides interacting with the inactive CLPXP that are from polypeptides from outside complex I? The data here seem a bit hard to interpret without some idea of the level of background binding.

This information is taken from the previously published screen for CLPP-substrates where all the technical details and information on additional substrates are provided (Szczepanowska et al, 2016). In brief, we have identified 66 potential CLPP substrates using a by taking advantage of the ability of inactivated CLPP to accept and retain proteins translocated into its chamber. We used catalytically inactive variant of CLPP to trap putative substrates, while expression of WT CLPP was used as a control. In the published list, beside Complex I subunit, we identified subunits of some other complexes and a number of proteins involved in mitochondrial gene expression, including processing and translation. So far, we characterized the role of ERAL1, a 12S rRNA chaperone and showed that CLPP activity is needed for the formation of functional mitochondrial ribosome by timely removal of ERAL1 from 28S small ribosomal subunit (Szczepanowska et al, 2016).

As this information was already published, we decided to take the table with peptide enrichment from Figure 1b out of the revised manuscript and leave only the figure marking the exact position of these subunits on CI structure.

5 In Fig 1c, are there control complex I polypeptides that were not affected by CLPP knock out that were not in the N module?

Indeed, this is an important control that we repeatedly tried to perform in several pulse-chase IPs for non-substrate CI subunits belonging to the different CI modules (NDUFS3, NDUF9; NDUF6). However, due to:

- (a) low efficiency of immunoprecipitation using the antibodies against the endogenous CI polypeptides that we have in our collection;
- (b) and/or the low turnover rates of the non-N-module subunits;

we were not able to obtain signals that would be convincing to support any definitive conclusions in this regard. As the quality of this data was not meeting our standards, we decided not to include them in the manuscript. Instead, we used Pulse SILAC Complexomics as a less biased and more sensitive approach. This analysis that fully supported our previous observations (Figure 3c and Extended data Table 5).

6. I don't follow how shear stress would be damaging to complex I and whether this is associated with loss of FMN content? Is it possible to assess FMN and FeS content of the free N modules? If the N module did contain an FMN I might expect this to produce ROS?

Our results strongly suggest that the N-module needs to be replaced at such a high rate not because it is more rapidly oxidized, but because it suffers shear stress and is "mechanically" rather instable. This matches the observation that the subunits with the highest exchange rate (as discussed under point 3.) seem to form some kind of glue between the N- and Q-module. This also fits nicely with our experiment showing that although in steady-state conditions, in the absence of CLPP N-module is not replaced, when challenged with stress it falls off more easily (Figure 4e). We do not know if this is directly connected to the loss of FMN content.

FeS content in the free module: We have used the NDUF11/CLPP DKO cells to analyze this, as they do not form CI, but retain the free N-module variants. In the manuscript we already included indirect measurement of the FeS content by performing the inverse redox shift assays (Figure 4d). As substantial number of cysteines in all three subunits are involved in the coordination of FeS clusters (see Figure R2, below), we propose that free N-modules that were never incorporated into CI do not contain FeS clusters, as cysteines in these subunits were accessible to NEM, a chemical that binds free thiols (Figure 4d).

FMN content in free module and intact CI:

Measurement of FMN content is not an easy task, as beside CI, a number of enzymes residing inside mitochondria contain this cofactor. We have used flavin fluorescence to assess the amount of FMN in intact mitochondria and in alamethicin-permeabilized mitochondrial membranes of wild type and CLPP deficient MEFs (Figure R1), as this has been successfully used to assess CI FMN (Stepanova et al, 2019). FMN content in free N-module is measured by using NDUFB11/CLPP (DKO) cells, while the NDUFB11 KO cells that do not have CI, nor they accumulate N-module, were used as negative control (Figure R1 and Fig 4f in the revised manuscript).

This analysis showed that CLPP deficient mitochondria indeed have lower FMN content when compared to controls. Mitochondrial permeabilization significantly decreased the levels of FMN in wild type mitochondria, suggesting that soluble FMN fraction contributes to the FMN pool upon normal conditions. Intriguingly, although the FMN levels were significantly decreased in intact CLPP deficient mitochondria, they remained unchanged upon the mitochondrial permeabilization suggesting that in CLPP KO mitochondria FMN resides primarily in membrane bound fraction. Changes in the FMN content between intact mitochondria and isolated mitochondrial membranes might also reflect levels of other flavin-containing proteins.

Figure R1. FMN content in wild type (WT), CLPP deficient (CLPP), NDUFB11 deficient (B11) and double NDUFB11/CLPP deficient cells (DKO) measured as flavin fluorescence. WT and CLPP cells were treated with MitoPQ that produces ROS specifically at FMN site of CI.

The treatment with mitoPQ, which was previously shown to stimulate the superoxide production selectively through the FMN site of CI (Robb et al, 2015), led to a severe loss of FMN from CLPP deficient mitochondria, while the FMN content in wild type mitochondria was very mildly affected, and only in intact mitochondria. This result indicates that CLPP-mediated CI salvage pathway can considerably compensate for a loss of essential FMN cofactor caused by extensive ROS production.

Finally, we did not observe any difference between the FMN content in NDUFB11 KO and CLPP/NDUFB11 DKO mitochondria, suggesting that free N-accumulation upon CLPP deficiency does not contain FMN cofactor, that might be only be added on the fully formed CI. This also means that the free N-module that accumulates in CLPP deficient cells is non-active and thus cannot produce ROS, in agreement with our results in Extended Data Figure 5 in the manuscript. However, this result must be interpreted with caution as the obtained concentrations in this case were at the lower limit of detection.

7 It should be possible to roughly assign the cys residues that are seen in Fig 4 to FeS centres, for example by considering the number of FeS numbers in this subunit and by calculating the stoichiometry of the Cys residues modified with the large molecular weight label

As mentioned earlier, substantial number of cysteine residues in all three core N-module subunits is involved in a formation of Fe-S clusters (NDUFV1: 33%; NDUFV2:66%; NDUFS1: 61%)(Figure R2A, red blocks).

Figure R2. Cysteine positions in core N-module subunits. (A) Positions of cysteines in core N-module subunits. (B) Positions of cysteines in Cryo-EM structure of N-module part of CI; (C) Close-up of the N1a FeS cluster and adjacent cysteines in Vi and V2 subunits. **red**-cysteines bound in FeS clusters; **orange** and **pink** - cysteines possibly involved in formation of disulfide bonds; **yellow** - cysteines likely not involved in disulfide bond formations.

Those cysteines are enclosed inside the catalytic core of N-module (Figure R2B-C, red spheres). Beside the cysteines that are directly engaged in a formation of FeS centers, N-module core possesses a substantial number of other cysteines. Pairs of cysteines belonging to NDUFV2 (orange) and NDUFV1 (pink) subunits are clustered in a close proximity to N1a FeS cluster that does not participate directly in the electron transfer (Figure R2C). Intriguingly, according to CI cryo-EM structure, these cysteines might form the intermolecular disulfide bonds that permit formation of the stable NDUFV1/NDUFV2 subcomplex (Figure R2C). ClpXP might facilitate the formation of disulfide bond between NDUFV1/NDUFV2 that likely occurs in parallel to N-module assembly into the rest of CI. This mechanism could explain why the free N-module subunits that accumulate in NDUFB11/CLPP DKO are mostly found as reduced forms (Figure 4d).

I may have missed something, but I wasn't clear how the authors could distinguish cys labelling of de novo synthesised subunits that were labelled before insertion of FeS centers, from damaged subunits that were removed from the complex with the cys residues becoming exposed as the FeS centers were stripped out?

As pointed out earlier, for this experiment we have used the *Ndufb11/Clpp* double knock out cell lines. NDUFB11 is a P-module subunit, whose loss disables formation of the CI (Stourd et al. 2016). Therefore, in the double mutant we were following only subunits that were never assembled into CI. In these conditions, we showed that cysteines were fully accessible to NEM binding, indication that they were not bound in FeS clusters (Figure 4d). This also indicates that free N-module that is accumulating in CLPP deficient cells, and is shown to be originating from newly formed proteins (Figure 3c-e), does not contain FeS clusters. We have now added a scheme in Figure 4d that should enable readers to better understand the reasoning behind this experiment and its conclusions.

8. The role of stalling of oxidative phosphorylation was unclear to me. Does stalling just mean slow respiration? Does this mean lowered respiration and a build-up of electrons in the respiratory chain? Does this imply that complex I is affected when electrons backed up produce ROS by reduction of the FMN on complex I, leading to damage to complex I? Is there any link between this process and the active/deactive transition in complex I? Does this imply that complex I is affected when electrons backed up produce ROS by reduction of the FMN on complex I, leading to damage to complex I? Is there any link between this process and the active/deactive transition in complex I?

Although our data do not allow us to make definite answers to this question we favor the hypothesis that a pool of the inactive CI cannot fulfill its NADH-oxidizing role anymore, therefore the overall oxidative phosphorylation is decreased. We think that in CLPP deficient cells we have a mixture of fully active CI that

contribute to oxidative phosphorylation, and inactivated CI units that cannot contribute to the oxidative phosphorylation any more. Therefore, the overall decrease in respiration, even in the presence of normal, if not slightly increased CI levels, comes from portion of CI that cannot contribute to proton pumping and formation of proton gradient. Nevertheless, we cannot exclude that N-module damage, which is normally removed by ClpXP might still modulate the CI activity. The overall decrease of ROS production under normal condition and upon increased stress (Extended Date Figure 3) in CLPP deficient cells would favor the first model. This model also suggests that N-module inactivation and subsequent degradation by ClpXP could play a role in preventing RET, for example during the ischaemia, while allowing quick restoration of CI function after the reperfusion event without a need to rebuild the entire complex. Intriguingly, one of our CI candidate substrates, NDUFS2 subunit, plays a direct and essential role in active/deactive transition of CI. Hence the role of ClpXP in a modulation or regulation of CI active/deactive transition remains feasible, but needs further experimental support.

9. Does the effect of CLPXP loss on zebra fish with unstable complex I correlate with an increase in complex I activity in isolated mitochondria? Or to changes in ATP/ADP ratio in the adult fish? Without some bioenergetic assessment I find it difficult to interpret these findings.

In this manuscript we have used *C. elegans* (not zebrafish) model of CI deficiency to test the possible beneficial effect of CLPP depletion. As this mutant has a specific CI mutation, we believe that the beneficial effect does not stem from the increased CI activity, but rather from its increased stability. Indeed, we do not detect difference in respiration rate that is very low regardless of CLPP levels (included in revised Figure S6E). Instead we speculate that stabilization of CI leads to higher levels of supercomplexes and this might have beneficial effect on the cristae morphology, hence mitochondrial wellbeing. Alternatively, presence of functional free N-module might provide means to maintain NAD⁺/NADH ratio when CI is not functioning properly. We have included more explanation for this in the revised manuscript.

Reviewer 4:

Szczepanowska have evaluated complex I assembly. They have demonstrated that the mitochondrial protease ClpP degrades complex I subunits in order to promote optimal complex I assembly into superstructures. The work involves intricate biochemical analysis and provides additional insights into the assembly of complex I and the function of ClpP. The paper is very dense and the methods are not fully explained. For example, the authors conduct complexome profiling. The authors cite a previous paper regarding this assay, but provide no description of the assay in the results or methods. This example and many others make the paper difficult to

understand and follow. In many places, the sentences are not well linked, so data is hard to understand. Often figures are cited without an explanation of the results. As such, I found the paper to be highly specialized. Perhaps with better writing it might be more accessible to a broader audience.

We thank the reviewer for the overall positive evaluation of our manuscript. We recognize that the manuscript is very dense with data and might be hard to follow at times. We have rewritten part of it in the revised manuscript. Also, we included detailed description of all methods (including complexome analysis).

The authors use ClpP knockouts and knockdowns, but don't show levels of ClpP in these cells. These data would be important controls to demonstrate changes in target levels. The authors use siRNA to knockdown targets like ClpX. In all of these cases, the immunoblot should be shown to demonstrate target knockdown. In addition, more than 1 siRNA or a rescue experiment should be performed to support on-target effects.

Many relevant controls of CLPP or CLPX levels were already included in the respective supplementary panels. In our experiments we used almost heart samples or isolated MEFs from whole body CLPP knockout mice that we have previously published (Szczepanowska et al 2016; Seiferling et al. 2016). In this manuscript we have already included control blots demonstrating complete loss of CLPP in heart tissues (Extended data Fig 1d); MEFs (Extended Data Fig 2b and 3b), HeLa cells (Extended Data Fig 2b and 3b) and NDUFB11/CLPP DKO (Extended data Fig 2b).

We have also performed a number of knockdowns and the control experiments showing clear down regulation of target proteins. Some of these experiments were already included in the manuscript: (1) CLPX knockdown in wild type MEFs (Extended data Fig. 3b) and (2) CLPX knockdown in mtDNA mutator MEFs (Extended data Fig. 6a). Now, we have included additional panel for: (3) knockdown of CLPX in COX10 KO cells (Extended data Fig. 6b);

(4) knockdown of CLPX in CYTB KO cells (Extended data Fig. 6b);

(5) knockdown of CLPX in control and CLPP KO cells (Fig. 4a);

(6) knockdown of LONP1 in control and CLPP KO cells;

For *Clpx* knockdown that we used repeatedly in the manuscript we tested two siRNAs that showed similar, very strong effect as shown on Extended data 3b and 6a. Therefore, in most of our other experiments we used only one siRNA to be able to handle multiple samples and very large quantities of cells used from mitochondrial isolation, that most of our experiments required. Exactly the same siRNAs were used previously in the knockdown experiments where we showed that other CLPP-substrates (ERAL1 and EFG1) accumulate to the same levels in CLPP knockout cells as in CLPX knockdown wild-type cells; while the upregulation stays the same upon loss of both CLPP and CLPX (Szczepanowska et al 2016).

For *Lonp1* we have also tested two siRNAs, of which, only one induced LONP1 depletion and was used in subsequent experiments (see above).

We thank once again all reviewers for their helpful insight and we hope that they find our revised manuscript clearer and suitable for publication in Nature Communication.

Sincerely,
Aleksandra Trifunovic

REVIEWERS' COMMENTS:

Reviewer #3 (Remarks to the Author):

I think that the authors have responded well to the many points I raised in my first review. As expected, many of these questions are not yet resolved experimentally, but the rewritten paper puts the areas of uncertainty in context which is important for those who will be following up this work. I support publication of this important paper.

Reviewer #4 (Remarks to the Author):

Many of my original comments related to experimental techniques have been addressed. However the paper remains dense and highly specialized.